# Optimism Without Regularization:
# Constant Regret in Zero-Sum Games

**John Lazarsfeld**
SUTD[*]

**Georgios Piliouras**
SUTD

**Ryann Sim**
SUTD

**Stratis Skoulakis**
Aarhus University

## Abstract

This paper studies the *optimistic* variant of Fictitious Play for learning in two-player zero-sum games. While it is known that Optimistic FTRL – a *regularized* algorithm with a bounded stepsize parameter – obtains constant regret in this setting, we show for the first time that similar, optimal rates are also achievable *without* regularization: we prove for two-strategy games that Optimistic Fictitious Play (using *any* tiebreaking rule) obtains only *constant regret*, providing surprising new evidence on the ability of *non*-no-regret algorithms for fast learning in games. Our proof technique leverages a geometric view of Optimistic Fictitious Play in the dual space of payoff vectors, where we show a certain energy function of the iterates remains bounded over time. Additionally, we also prove a regret *lower bound* of $\Omega(\sqrt{T})$ for *Alternating* Fictitious Play. In the unregularized regime, this separates the ability of optimism and alternation in achieving $o(\sqrt{T})$ regret.

## 1 Introduction

Despite the fact that regularization is essential for no-regret online learning in general adversarial settings, *un*regularized algorithms are still able to obtain *sublinear regret* in the case of two-player zero-sum games. Fictitious Play (FP), dating back to Brown (1951), is the canonical example of such an unregularized algorithm, and it results from both players independently running Follow-the-Leader (FTL).[2] In the worst case, FTL can have $\Omega(T)$ regret due to its sensitivity to oscillations in adversarially-chosen reward sequences (Shalev-Shwartz et al., 2012). However, in zero-sum game settings, the classic result of Robinson (1951) proved that, under Fictitious Play, the *sum* of the players' regrets (henceforth referred to as *regret*) is indeed sublinear (thus implying time-average convergence to a Nash equilibrium), albeit at the very slow $O(T^{1-1/n})$ rate for $n \times n$ games (which was shown to be tight by Daskalakis and Pan (2014) using an adversarial tiebreaking rule).

However, the recent works of Abernethy et al. (2021a) and Lazarsfeld et al. (2025) have established improved $O(\sqrt{T})$ regret guarantees for Fictitious Play on diagonal payoff matrices (using lexicographical tiebreaking) and on generalized Rock-Paper-Scissors matrices (using any tiebreaking rule), respectively. As a result, there is growing evidence on the robustness of unregularized algorithms like FP (*not* a no-regret algorithm in general) for obtaining fast, sublinear regret in zero-sum games.

On the other hand, the past decade has seen *regularized* learning algorithms exhibit a remarkable success in providing even better $o(\sqrt{T})$ regret guarantees for learning in games. In zero-sum games, *optimistic* variants of FTRL (including Optimistic Multiplicative Weights and Optimistic Gradient Descent) obtain only *constant* regret (with respect to the time horizon $T$), implying optimal $O(1/T)$ time-average convergence to Nash (Rakhlin and Sridharan, 2013; Syrgkanis et al., 2015). While such guarantees can be obtained using absolute constant stepsizes (with no dependence on $T$), standard

---

[*]Correspondence to: `jlazarsfeld@gmail.com`

[2]FTL is a particular instance of Follow-the-Regularized-Leader (FTRL) with unbounded stepsize $\eta \to \infty$.

proof techniques (e.g., the RVU bound approach of Syrgkanis et al. (2015)) still crucially require a *finite upper bound on stepsize*, corresponding to constant magnitudes of regularization. This raises the following, fundamental question: *Is $O(1)$ regret attainable in zero-sum games without regularization (equivalently, with unbounded stepsizes)? Can variants of Fictitious Play achieve $O(1)$ regret?*

Apart from its theoretical interest, this question admits crucial applications in the context of equilibrium computation algorithms for combinatorial games (Beaglehole et al., 2023), as well as during training via self-play in certain multi-agent reinforcement learning settings (Vinyals et al., 2019).

## 1.1 Our Contributions

In this work, we establish an affirmative answer to the question above. Our main result establishes that, in the case of $2\times2$ zero-sum games, *Optimistic Fictitious Play* obtains constant regret:

**Informal Theorem** (See Theorem 3.1). *Optimistic Fictitious Play, using any tiebreaking rule, obtains $O(1)$ regret in all $2\times2$ zero-sum games with a unique, interior Nash equilibrium.*

Our result gives surprising new evidence that, even without regularization, optimism can be used to obtain an accelerated regret bound, matching the optimal rate obtained by regularized Optimistic FTRL algorithms (Syrgkanis et al., 2015). While our theorem establishes constant regret only for the case of two-strategy zero-sum games, our proof techniques offer indication that similar, optimal regret bounds may further hold in higher-dimensional settings. Apart from our theoretical results, we also experimentally evaluate Optimistic FP on higher-dimensional zero-sum games, and these evaluations suggest that, even for much larger games, Optimistic FP still obtains constant regret.

Our proof technique is based on a novel geometric perspective of Optimistic FP in the dual space of payoff vectors. Our main technical contribution is showing that an *energy function* of the dual iterates of the algorithm is upper bounded by a constant. This energy upper bound can then be easily used to establish constant regret of the primal iterates. The latter comes in contrast to the energy growth of the iterates of standard FP, which strictly increases over the time horizon (Lazarsfeld et al., 2025).

We also consider the *alternating* variant of Fictitious Play. Recent work has studied the use of alternation (independently of optimism) as a method for obtaining $o(\sqrt{T})$ regret guarantees in both the adversarial (Gidel et al., 2019; Bailey et al., 2020; Cevher et al., 2023; Hait et al., 2025) and the zero-sum game setting (Wibisono et al., 2022; Katona et al., 2024). Contrary to *optimism*, we show in the case of *alternation* that regularization is necessary to achieve $o(\sqrt{T})$ regret:

**Informal Theorem** (See Theorem 3.2). *On the $2\times2$ Matching Pennies game, Alternating Fictitious Play, using any tiebreaking rule and for nearly all initializations, has regret at least $\Omega(\sqrt{T})$.*

Together, our results separate the regret guarantees of using optimism and alternation in the regime of no regularization: while optimism without regularization can obtain optimal $O(1)$ regret (Theorem 3.1), alternation alone is in general insufficient for improving beyond $O(\sqrt{T})$ (Theorem 3.2), the same rate achievable by standard (non-alternating) FP in the $2\times2$ setting. To this latter point, note that the lower bound of Theorem 3.2 comes in contrast to the improved $O(T^{1/5})$ rate obtainable by Alternating FTRL under a sufficiently small stepsize (Katona et al., 2024).

Table 1 summarizes our results and the landscape of regret guarantees in the $2\times2$ setting for FTRL and FP variants, and Figure 1 shows an example of the empirical regret guarantees of standard, Optimistic, and Alternating FP variants in several games (additional results are presented in Sections 5 and E).

|  | Standard | Optimistic | Alternating |
|---|---|---|---|
| $\eta$ bounded (FTRL) | $O(\sqrt{T})$ † | $O(1)$ ^ | $O(T^{1/5})$ ^^ |
| $\eta \to \infty$ (FP) | $O(\sqrt{T})$ ‡ | $O(1)$ ⋆ | $\Omega(\sqrt{T})$ ⋆⋆ |

Table 1: Regret guarantees for FTRL and Fictitious Play variants in $2\times2$ zero-sum games, with our contributions shaded in gray. †: Using the standard setting of $\eta = 1/\sqrt{T}$ (Shalev-Shwartz et al., 2012). ^: Via the RVU bounds of Syrgkanis et al. (2015). ^^: (Katona et al., 2024), extending on the prior $O(T^{1/3})$ bound of Wibisono et al. (2022). ‡: Implicit in the proof of Robinson (1951). ⋆: Theorem 3.1. ⋆⋆: Theorem 3.2.

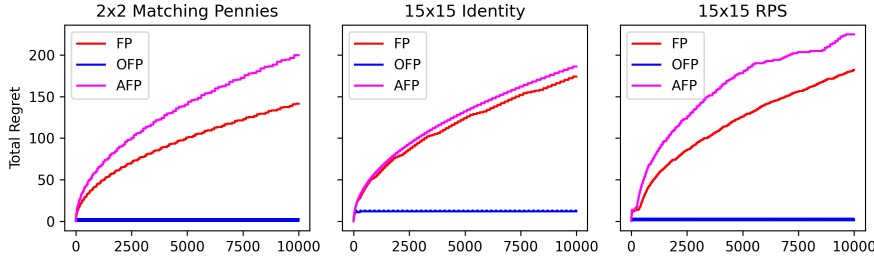

Figure 1: Empirical regret of standard (FP), Optimistic (OFP), and Alternating (AFP) Fictitious Play in Matching Pennies (from $x_1^0 = (1/3, 2/3)$, $x_2^0 = (2/3, 1/3)$), on the 15×15 identity matrix (from $x_1^0 = e_1$, $x_2^0 = e_n$), and on 15×15 generalized Rock-Paper-Scissors (from $x_1^0 = e_1$, $x_2^0 = e_n$). Each algorithm was run for $T = 10000$ iterations using a lexicographical tiebreaking rule. Each subfigure demonstrates the constant empirical regret of OFP compared to the roughly $\sqrt{T}$ regret growth of standard FP and AFP. More experimental details and results are given in Section 5 and Section E.

## 1.2 Other Related Work

**Optimistic learning in games.** Our work relates to a line of research on the convergence properties of optimistic and extragradient-type algorithms in both normal-form and extensive-form games (Daskalakis et al., 2021; Fasoulakis et al., 2022; Anagnostides et al., 2022a; Zhang et al., 2024; Farina et al., 2022; Hsieh et al., 2022; Anagnostides et al., 2022b; Piliouras et al., 2022; Anagnostides et al., 2022c). As previously noted, a key difference is that these approaches typically rely on constant or time-decreasing step sizes, corresponding to some level of regularization. Recent works have also investigated last-iterate convergence properties of optimistic and extragradient methods (Daskalakis and Panageas, 2019; Cai et al., 2022; Abernethy et al., 2021b; Hsieh et al., 2020), as well as for variants of regret-matching, including under alternation (Cai et al., 2025a). Other works have studied accelerated rates using optimism *without* regularization in certain Frank-Wolfe-type, convex-concave saddle-point problems (Wang and Abernethy, 2018; Abernethy et al., 2018).

**Learning in 2×2 games.** Our work adds to a growing recent literature studying online learning algorithms in 2×2 games: in two-strategy zero-sum games, Bailey and Piliouras (2019) proved that Online Gradient Descent obtains $O(\sqrt{T})$ regret even with large constant stepsizes. More recent works of Cai et al. (2024) and Cai et al. (2025b) establish lower-bounds on the last-iterate and random-iterate convergence rates of Optimistic MWU using a hard 2×2 construction, as well as an $O(T^{1/6})$ upper bound on best-iterate convergence for 2×2 zero-sum games. Chen and Peng (2020) similarly used a 2×2 construction to establish a general $\Omega(\sqrt{T})$ lower bound on the regret of standard Multiplicative Weights. In a family of two-strategy congestion games, Chotibut et al. (2021) also showed that the iterates of Multiplicative Weights can exhibit formally chaotic behavior.

**Fictitious Play.** We refer the reader to the recent results of Daskalakis and Pan (2014), Abernethy et al. (2021a), and Lazarsfeld et al. (2025) (and the references therein) for background on standard Fictitious Play. Abernethy et al. (2021a) also briefly introduced the optimistic variant of Fictitious Play, and they informally conjecture the algorithm to have constant regret on diagonal payoff matrices. The convergence behavior of FP has also been studied in potential games (Monderer and Shapley, 1996; Panageas et al., 2023), near-potential games (Candogan et al., 2013), Markov games (Sayin et al., 2022; Baudin and Laraki, 2022), and extensive-form games (Heinrich et al., 2015).

## 2 Preliminaries

Let $[n] = \{1, \ldots, n\}$, let $\Delta_n$ be the probability simplex in $\mathbb{R}^n$, and let $\{e_i\}_n = \{e_i : i \in [n]\} \subset \Delta_n$ denote the set of standard basis vectors in $\mathbb{R}^n$, which correspond to vertices of $\Delta_n$. For $x \in \Delta_n$, we say $x$ is *interior* if $x_i > 0$ for all $i \in [n]$.

### 2.1 Online Learning in Zero-Sum Games

Let $A \in \mathbb{R}^{m \times n}$ be the payoff matrix for a two-player zero-sum game, and let $T$ be a fixed time horizon. At round $t$, Players 1 and 2 simultaneously choose mixed strategies $x_1^t \in \Delta_m$ and $x_2^t \in \Delta_n$, obtain payoffs $\langle x_1^t, A x_2^t \rangle$ and $-\langle x_2^t, A^\top x_1^t \rangle$, and observe feedback $A x_2^t$ and $-A^\top x_1^t$, respectively.

**Regret and Convergence to Nash.** Each player seeks to maximize their cumulative payoff, and their performance is measured by the individual regrets $\text{Reg}_1(T) = \max_{x \in \Delta_m} \sum_{t=1}^{T} \langle x - x_1^t, A x_2^t \rangle$ and $\text{Reg}_2(T) = \min_{x \in \Delta_n} \sum_{t=1}^{T} \langle x_2^t - x, A^\top x_1^t \rangle$. From a global perspective, we study the *total regret* (henceforth *regret*) $\text{Reg}(T) = \text{Reg}_1(T) + \text{Reg}_2(T)$ given by

$$\text{Reg}(T) = \max_{x \in \Delta_m} \left\langle x, \sum_{t=0}^{T} A x_2^t \right\rangle - \min_{x \in \Delta_n} \left\langle x, \sum_{t=0}^{T} A^\top x_1^t \right\rangle . \tag{1}$$

It is well known that sublinear bounds on $\text{Reg}(T)$ correspond to convergence (in duality gap) of the players' time-average iterates to a Nash equilibrium (NE) of $A$. Recall that the duality gap of a joint strategy profile $(x_1, x_2)$ is given by $\text{DG}(x_1, x_2) = \max_{x_1' \in \Delta_m} \langle x_1', A x_2 \rangle - \min_{x_2' \in \Delta_n} \langle x_2', A^\top x_1 \rangle$, and that $(x_1^*, x_2^*)$ is an NE of $A$ if and only if $\text{DG}(x_1^*, x_2^*) = 0$. Then the following relationship holds (see Section A for a proof):

**Proposition 2.1.** *Let $\widetilde{x}_1^T = \frac{1}{T}(\sum_{t=0}^{T} x_1^t)$ and $\widetilde{x}_2^T = \frac{1}{T}(\sum_{t=0}^{T} x_2^t)$ denote the time-average iterates of Players 1 and 2, respectively, and suppose $\text{Reg}(T) = o(T)$. Then $(\widetilde{x}_1^T, \widetilde{x}_2^T)$ converges (in duality-gap) to an NE of $A$ at a rate of $\text{Reg}(T)/T = o(1)$.*

## 2.2 Fictitious Play and Optimistic Fictitious Play

We now introduce the Optimistic Fictitious Play (and standard Fictitious Play) algorithms. The primal update rules for both standard and Optimistic Fictitious Play can be described via the following $\alpha$-Optimistic Fictitious Play ($\alpha$-OFP) expression:

$$
\begin{aligned}
x_1^{t+1} &:= \operatorname*{argmax}_{x \in \{e_i\}_m} \left\langle x, \sum_{k=0}^{t} A x_2^k + \alpha A x_2^t \right\rangle \\
x_2^{t+1} &:= \operatorname*{argmax}_{x \in \{e_i\}_n} \left\langle x, \sum_{k=0}^{t} -A^\top x_1^k - \alpha A^\top x_1^t \right\rangle .
\end{aligned}
\tag{$\alpha$-OFP}
$$

When $\alpha = 0$, then ($\alpha$-OFP) recovers standard Fictitious Play, where each player's strategy at time $t + 1$ is a best response to the sum of its feedback vectors through round $t$. Optimistic Fictitious Play (OFP) is the setting of ($\alpha$-OFP) with $\alpha = 1$. Observe this recovers the *unregularized variant* of Optimistic FTRL (equivalently, with $\eta \to \infty$) in the zero-sum game setting (c.f., (Rakhlin and Sridharan, 2013; Syrgkanis et al., 2015)), which adds bias to the most recent feedback vector.

**Remark 2.2 (Tiebreaking Rules).** Observe that the argmax sets in ($\alpha$-OFP) may contain multiple vertices. For this, we assume that the argmax operator encodes a *tiebreaking rule* that returns a distinct element. Throughout, we make *no assumptions on the nature of the tiebreaking rule*, and in general ties can be broken deterministically, randomly, or adaptively/adversarially.

**Dual payoff vectors and primal-dual update.** Optimistic FP can be equivalently written with respect to the *cumulative* payoff vectors $y_1^t = \sum_{k=0}^{t-1} A x_2^k \in \mathbb{R}^m$ and $y_2^t = \sum_{k=0}^{t-1} -A^\top x_1^k \in \mathbb{R}^n$. Specifically, the iterates of the algorithm can be expressed in the following *primal-dual* form:

**Definition 2.3.** Let $y_1^0 = 0 \in \mathbb{R}^m$ and $y_2^0 = 0 \in \mathbb{R}^n$, and fix any initial $x_1^0 \in \Delta_m$ and $x_2^0 \in \Delta_n$. Then for $t \geq 1$, the dual (i.e., $(y_1^t, y_2^t)$) and primal (i.e., $(x_1^t, x_2^t)$) iterates of Optimistic FP are:

$$
\begin{cases}
y_1^t = y_1^{t-1} + A x_2^{t-1} \\
y_2^t = y_2^{t-1} - A^\top x_1^{t-1}
\end{cases}
\quad \text{and} \quad
\begin{cases}
x_1^t = \operatorname*{argmax}_{x \in \{e_i\}_m} \left\langle x, y_1^t + A x_2^{t-1} \right\rangle \\
x_2^t = \operatorname*{argmax}_{x \in \{e_i\}_n} \left\langle x, y_2^t - A^\top x_1^{t-1} \right\rangle
\end{cases}
. \tag{OFP}
$$

# 3 Regret Bounds for Optimistic and Alternating Fictitious Play

The main result of this paper establishes a *constant* regret bound for Optimistic Fictitious Play in two-strategy zero-sum games. Formally we prove the following theorem:

**Theorem 3.1.** *Let $A$ be a 2x2 zero-sum game with a unique interior NE, and let $\{(x_1^t, x_2^t)\}$ be the iterates of (OFP) on $A$ using any tiebreaking rule. Then $\text{Reg}(T) \leq O(1)$.*

As mentioned, this result establishes the first constant regret bounds for Fictitious Play variants in the two-player zero-sum game setting, and the result holds regardless of the tiebreaking rule used.

Moreover, this bound matches the optimal rate obtained by Optimistic FTRL variants for zero-sum games (Syrgkanis et al., 2015), but notably, our proof technique departs significantly from the RVU bound approach used to obtain those results.

As a consequence of the techniques we develop for proving Theorem 3.1, we also establish a *lower bound* on the regret of *Alternating* Fictitious Play. In particular, for the corresponding *alternating regret* $\text{Reg}^{\text{alt}}(T)$ (see Definition D.2), we prove on the $2{\times}2$ Matching Pennies game the following:

**Theorem 3.2.** *Suppose $x_1^1 = (p, 1-p) \in \Delta_2$ for irrational $p \in (3/4, 1)$, and let $\{x^t\}$ be the iterates of Alternating FP on* (Matching Pennies) *using any tiebreaking rule. Then $\text{Reg}^{\text{alt}}(T) \geq \Omega(\sqrt{T})$.*

To streamline the presentation of the paper, we defer the precise descriptions of the alternating play model, alternating regret, and the Alternating FP algorithm to Section D, where we also develop the proof of Theorem 3.2. Instead, in the remainder of the main text, we focus on developing the proof of Threorem 3.1. To this end, we proceed to give an overview of our techniques.

## 3.1 Intuition and Overview of Proof Techniques for Theorem 3.1

To prove Theorem 3.1, we leverage a geometric view of Optimistic Fictitious Play in the dual space of payoff vectors. We give a brief overview of this geometric perspective here:

**Energy and regret.**  First, we show that the regret of the primal iterates $\{x^t\}$ is equivalent to the growth of an *energy function* of the dual iterates $\{y^t\}$. Specifically, define the energy $\Psi$ as follows:

**Definition 3.3.** Let $y^t := (y_1^t, y_2^t)$ be the concatenated primal and dual iterates of (OFP) at time $t \geq 1$. Then for $y = (y_1, y_2) \in \mathbb{R}^{m+n}$, the energy function $\Psi : \mathbb{R}^{m+n} \to \mathbb{R}$ is given by

$$\Psi(y) \;=\; \max_{x \in \Delta_m \times \Delta_n} \langle x, y \rangle . \tag{2}$$

In other words, $\Psi$ is the *support function* of $\Delta_m \times \Delta_n$. Then by definition of $\text{Reg}(T)$ and the payoff vectors $\{y^t\}$, the following relationship holds (see Section A for a proof):

**Proposition 3.4.** *Let $\{x^t\}$ and $\{y^t\}$ be the iterates of* (OFP). *Then $\text{Reg}(T) = \Psi(y^{T+1})$.*

Due to Proposition 3.4, it is immediate that a constant upper bound on $\Psi(y^{T+1})$ implies a constant upper bound on $\text{Reg}(T)$. To this end, our main technical contribution is to prove the following upper bound on the energy $\Psi$ under Optimistic Fictitious Play:

**Theorem 3.5.** *Assume the setting of Theorem 3.1. Let $a_{\max} = \|A\|_\infty$ denote the largest entry of $A$, and let $a_{\text{gap}} = \min_{(i,j),(k,\ell)} |A_{ij} - A_{k\ell}|$ denote the smallest absolute difference between two entries of $A$. Let $\{y^t\}$ denote the dual iterates of* (OFP) *on $A$. Then $\Psi(y^{T+1}) \leq 8a_{\max}\big(1 + 2\big(\frac{a_{\max}}{a_{\text{gap}}}\big)\big)^2$.*

In other words, the energy of the dual iterates under Optimistic FP are bounded by an absolute constant that depends only on the entries of $A$. In Section 4, we give a technical overview of the proof of Theorem 3.5, but we first present more introduction and intuition on the geometric perspective of Optimistic Fictitious Play that is used to prove the result.

**Fictitious Play as Skew-Gradient Descent.**  As shown in Abernethy et al. (2021a) and Lazarsfeld et al. (2025), in the dual space of payoff vectors, standard Fictitious Play can be viewed as a certain *skew-gradient descent* with respect to the energy $\Psi$. In light of this, we introduce a common geometric viewpoint that captures both standard and Optimistic FP and gives insight into their differences in energy growth, as implied by Theorem 3.5.

For this, note that the *subgradient set* of $\Psi$ at $y \in \mathbb{R}^{m+n}$ is given by $\partial\Psi(y) = \text{argmax}_{x \in \Delta_m \times \Delta_n} \langle x, y \rangle$. Then both standard FP and Optimistic FP can be expressed as a *skew-(sub)gradient-descent* with respect to $\Psi$ evaluated at a *predicted* dual vector $\widetilde{y}^{t+1}$ (see Section A.3 for a full derivation):

**Proposition 3.6.** *Let $\{y^t\}$ denote the dual iterates of either standard Fictitious Play (e.g., ($\alpha$-OFP) with $\alpha = 0$) or Optimistic Fictitious Play. Then for all $t \geq 1$, the iterates of each algorithm evolve as*

$$\begin{cases} y^t \;=\; y^{t-1} + Jx^{t-1} \\ x^t \;\in\; \partial\Psi\big(\widetilde{y}^{t+1}\big) \end{cases} \quad \text{where } J = \begin{pmatrix} 0 & A \\ -A^\top & 0 \end{pmatrix} \text{ and } \widetilde{y}^{t+1} = \begin{cases} y^t & \text{for FP} \\ 2y^t - y^{t-1} & \text{for OFP} \end{cases}, \tag{3}$$

*and it follows inductively that $y^{t+1} = y^t + J\partial\Psi(\widetilde{y}^{t+1})$, where $\partial\Psi(\widetilde{y}^{t+1})$ denotes a fixed vector in the subgradient set of $\Psi$ at $\widetilde{y}^{t+1}$.*

**One-step energy growth comparison of Fictitious Play variants.** For standard FP, due to its Hamiltonian structure, the analogous skew-gradient flow in continuous-time is known to exactly *conserve* $\Psi$, leading to constant regret (see, e.g., Mertikopoulos et al. (2018); Abernethy et al. (2021a); Wibisono et al. (2022)). However, due to discretization, this energy conservation does not hold in general under discrete-time Fictitious Play variants. For example, under each step of standard Fictitious Play, $\Psi$ is always non-decreasing. To see this, let $\Delta\Psi(y^t) = \Psi(y^{t+1}) - \Psi(y^t)$, and by slight abuse of notation, let $\partial\Psi(y)$ denote a fixed vector in the subgradient set of $\Psi$ at $y \in \mathbb{R}^{m+n}$. Then by Jensen's inequality and skew-symmetry of $J = -J^\top$, it holds for all $t \geq 1$ that

$$\textbf{For FP:} \quad \Delta\Psi(y^t) \;\geq\; \left\langle \partial\Psi(y^t), J\partial\Psi(\widetilde{y}^{t+1}) \right\rangle = \left\langle \partial\Psi(y^t), J\partial\Psi(y^t) \right\rangle = 0 \;. \tag{4}$$

In fact, the recent upper bounds of Abernethy et al. (2021a) and Lazarsfeld et al. (2025) imply that under FP, $\Psi$ is *strictly* increasing by a constant in roughly $\sqrt{T}$ iterations.

On the other hand, for Optimistic FP using $\widetilde{y}^{t+1} = 2y^t - y^{t-1}$, we instead have by Jensen's inequality:

$$\textbf{For Optimistic FP:} \quad \Delta\Psi(y^t) \leq \left\langle \partial\Psi(y^{t+1}), J\partial\Psi(\widetilde{y}^{t+1}) \right\rangle = \left\langle \partial\Psi(y^{t+1}), J\partial\Psi(\widetilde{y}^{t+1}) \right\rangle. \tag{5}$$

Thus by skew-symmetry of $J$, expression (5) reveals that for any $t$ where $\partial\Psi(y^{t+1}) = \partial\Psi(\widetilde{y}^{t+1})$ (e.g., true dual vector $y^{t+1}$ and predicted dual vector $\widetilde{y}^{t+1}$ both "map" to the same primal vertex), then the one-step energy growth $\Delta\Psi(y^t) \leq 0$ is *non-increasing* under Optimistic FP.

**Challenges in establishing non-positive energy growth for OFP.** Naively, one might in general hope the invariant $\partial\Psi(y^{t+1}) = \partial\Psi(\widetilde{y}^{t+1})$ holds at *every* timestep under Optimistic FP. However, simple experiments reveal that this is not true: in general $\Psi$ *can increase* during one step of the algorithm. Thus, understanding when and why such an invariant *does* hold is still a challenging task that may require leveraging structural properties of the payoff matrix. In the proof of Theorem 3.5, we leverage such properties of 2×2 games to establish sufficient conditions for when the above invariant holds, and this subsequently leads to a constant upper bound on energy.

## 4 Bounded Energy Under Optimistic Fictitious Play

In this section, we now give a technical overview of the proof of Theorem 3.5. Throughout the proof, we make the following assumptions on the payoff matrix $A$:

**Assumption 1.** Let $A \in \mathbb{R}^{2\times 2}$. Assume that

$$A \;=\; \begin{pmatrix} a & b \\ c & d \end{pmatrix} \quad \text{where} \quad \begin{cases} \text{(i)} & \det A = ad - bc = 0 \\ \text{(ii)} & a, d \;>\; \max\{0, b, c\} \end{cases} \;.$$

As proven by Bailey and Piliouras (2019), who studied online gradient descent in 2×2 games, for any Fictitious Play or FTRL variant, Assumption 1 holds without loss of generality:

**Proposition 4.1** (Bailey and Piliouras (2019))**.** *Let $A \in \mathbb{R}^{2\times 2}$ have a unique, interior NE, and let $\{x^t\}$ be the iterates of (OFP) on $A$. Then there exists $\widetilde{A} \in \mathbb{R}^{2\times 2}$ satisfying Assumption 1 such that (1) $\widetilde{A}$ and $A$ have the same NE and (2) the iterates $\{\widetilde{x}^t\}$ of running (OFP) on $\widetilde{A}$ are identical to $\{x^t\}$.*

For completeness, we include a full proof of this result in Proposition 4.1 of Section B. The key consequence of the assumption is that, under Optimistic FP, the dual payoff vectors $y^t = (y_1^t, y_2^t) \in \mathbb{R}^4$ all lie in the *same two-dimensional subspace*. Formally, we have:

**Proposition 4.2.** *Let $A$ satisfy Assumption 1, and let $\{y_1^t\}$ and $\{y_2^t\}$ be the dual payoff vectors of (OFP). Then for every $t \geq 1$, it holds that $y_{12}^t = -\rho_1 \cdot y_{11}^t$ and $y_{22}^t = -\rho_2 \cdot y_{21}^t$, where $\rho_1 := (d-c)/(a-b) > 0$ and $\rho_2 = (d-b)/(a-c) > 0$.*

The proof of Proposition 4.2 is given in Section C. Importantly, as $\rho_1, \rho_2 > 0$, observe that $y_{11}^t > 0 \iff y_{11}^t > y_{12}^t$, and $y_{21}^t > 0 \iff y_{21}^t > y_{22}^t$ for all times $t \geq 1$. Thus, in the 2×2 setting, the coordinates $y_{11}^t$ and $y_{21}^t$ encode all information needed to analyze the iterates of Optimistic FP in (3).

With this in mind, the strategy for proving the upper bound on the energy $\Psi(y^{T+1})$ is as follows: first, leveraging the observations above, and as in Bailey and Piliouras (2019), we restrict our study of the dual iterates $(y_1^t, y_2^t) \in \mathbb{R}^4$ to the pair of scalar iterates $(y_{11}^t, y_{21}^t) \in \mathbb{R}^2$. For this, we introduce in Section 4.1 a new set of notation to capture this lower-dimensional *subspace dynamics*, which also naturally leads to the definition of an *equivalent energy function*. For the new, equivalent energy function, we then prove in Section 4.2 a set of invariants that allow for establishing a uniform, constant upper bound on the energy of the dual iterates over time.

## 4.1 Subspace Dynamics of Optimistic Fictitious Play

We now introduce an equivalent set of primal and dual iterates $\{w^t\}$ and $\{z^t\}$ for (OFP), as well as a new, equivalent energy function $\psi$. We will establish in Proposition 4.6 that $\Psi(y^{T+1}) = \psi(z^{t+1})$.

**Primal variables.** First, by definition of (OFP), both $x_1^t$ and $x_2^t$ are vertices of $\Delta_2$. Letting $\mathcal{X} = \{(1,0),(0,1)\} \times \{(1,0),(0,1)\} \subset \mathbb{R}^4$ denote the vertices of the joint simplex $\Delta_2 \times \Delta_2$, it follows for each $t \geq 1$ that $x^t \in \mathcal{X}$. We define new primal iterates $w^t \in \mathbb{R}^4$, where each $w^t$ is a standard basis vector of $\mathbb{R}^4$. Let $\mathcal{W} = \{e_1, e_2, e_3, e_4\} \subset \mathbb{R}^4$ denote this set. Then for $t \geq 1$, let:

$$w^t = \begin{cases} e_2 \iff x^t = (0,1,1,0) & e_3 \iff x^t = (1,0,1,0) \\ e_1 \iff x^t = (0,1,0,1) & e_4 \iff x^t = (1,0,0,1) \end{cases}. \tag{6}$$

**Dual variables.** For each $t \geq 0$, let $z_1^t = y_{11}^t \in \mathbb{R}$ and $z_2^t = y_{21}^t \in \mathbb{R}$. Let $z^t = (z_1^t, z_2^t) \in \mathbb{R}^2$.

**Primal-dual map.** To aid in the description and analysis of the subspace dynamics, we describe the following partition of the dual space $\mathbb{R}^2$. We then describe a corresponding choice map $\mathsf{Q} : \mathbb{R}^2 \to \mathcal{X}$ that relates the primal and dual variables $\{w^t\}$ and $\{z^t\}$.

**Definition 4.3.** First, let $\mathcal{P} = \{P_1, P_2, P_3, P_4\} \subset \mathbb{R}^2$, where each $P_i$ is the set

$$\begin{aligned} P_2 &= \{z \in \mathbb{R}^2 : z_1 < 0 \text{ and } z_2 > 0\} & P_3 &= \{z \in \mathbb{R}^2 : z_1 > 0 \text{ and } z_2 > 0\} \\ P_1 &= \{z \in \mathbb{R}^2 : z_1 < 0 \text{ and } z_2 < 0\} & P_4 &= \{z \in \mathbb{R}^2 : z_1 > 0 \text{ and } z_2 < 0\}. \end{aligned} \tag{7}$$

Next, let $\widetilde{\mathcal{P}} = \{P_{1 \sim 2}, P_{2 \sim 3}, P_{3 \sim 4}, P_{4 \sim 1}\} \subset \mathbb{R}^2$, where we define

$$\begin{aligned} P_{2 \sim 3} &= \{z \in \mathbb{R}^2 : z_1 = 0 \text{ and } z_2 > 0\} & P_{3 \sim 4} &= \{z \in \mathbb{R}^2 : z_1 > 0 \text{ and } z_2 = 0\} \\ P_{1 \sim 2} &= \{z \in \mathbb{R}^2 : z_1 < 0 \text{ and } z_2 = 0\} & P_{4 \sim 1} &= \{z \in \mathbb{R}^2 : z_1 = 0 \text{ and } z_2 < 0\}. \end{aligned} \tag{8}$$

Finally let $\widehat{\mathcal{P}} = \cup_{i \in [4]} \widehat{P}_i$, where $\widehat{P}_i = P_i \cup P_{i \sim (i+1)}$. Observe by definition that $\widehat{\mathcal{P}} \cup \{(0,0)\} = \mathbb{R}^2$.

Note that for notational convenience, when using an index $i \in [4]$ in the context of the sets of Definition 4.3, we assume addition and subtraction to $i$ are performed (mod 4) in the natural way that maps to the set $\{1, 2, 3, 4\}$. For example, $P_{i \sim (i+1)}$ is the set $P_{4 \sim 1}$ when $i = 4$, and $P_{i-2} = P_{i+2}$ is the set $P_3$ when $i = 1$, etc. Figure 2 depicts the sets from Definition 4.3.

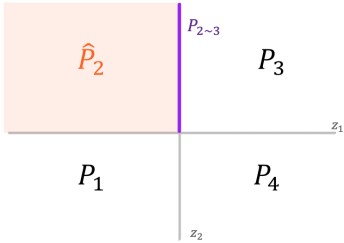

Figure 2: Examples of the sets in $\mathcal{P}$, $\widetilde{\mathcal{P}}$, and $\widehat{\mathcal{P}}$.

Moreover, Definition 4.3 allows for defining a choice map $\mathsf{Q} : \mathbb{R}^2 \to \mathcal{W}$ that encodes the primal update rule of (OFP). In particular, $\mathsf{Q}$ maps dual variables $z$ to primal vertices $w$ depending on the membership of $z$ in $\mathcal{P}$ or $\widetilde{\mathcal{P}}$. Formally:

**Definition 4.4.** Let $\mathsf{Q} : \mathbb{R}^2 \to \mathcal{W}$ be the map defined as follows: first, let $\mathsf{Q}((0,0)) = e_1$. Then

- For $z \in \mathcal{P}$: if $z \in P_i$ for $i \in [4]$, then $\mathsf{Q}(z) = e_i$.

- For $z \in \widetilde{\mathcal{P}}$: if $z \in P_{i \sim (i+1)}$ for $i \in [4]$, then $\mathsf{Q}(z) \in \{e_i, e_{i+1}\}$.

As in the full update rule of (OFP), we assume $\mathsf{Q}$ encodes a tiebreaking rule to ensure $\mathsf{Q}(z)$ returns a single element from $\mathcal{W}$. As in Remark 2.2, we make no assumptions on how such ties are broken.

**Primal-dual dynamics.** Using the definitions of the primal and dual variables $\{w^t\}$ and $\{z^t\}$ and the choice map $\mathsf{Q}$, the primal-dual dynamics of (OFP) can be rewritten as follows: for all $t \geq 2$, let

$$\begin{cases} z^t = z^{t-1} + Sw^{t-1} \\ w^t = \mathsf{Q}(z^t + Sw^{t-1}) \end{cases} \quad \text{where} \quad S = \begin{pmatrix} b & a & a & b \\ -c & -c & -a & -a \end{pmatrix}. \tag{9}$$

Observe that the $i$'th column of $S \in \mathbb{R}^{2 \times 4}$ are the entries $(\Delta y_{11}^t, \Delta y_{12}^t)$ when $w^t = e_i$ (cf., the definition of $w^t$ from expression (6)). Moreover, expression (9) implies $Sw^{t-1} = z^t - z^{t-1}$ for all

$t \geq 2$. Thus, we can further describe the primal-dual dynamics of (9) solely in terms of the sequence of dual variables $\{z^t\}$. In particular, for all $t \geq 2$, we have

$$\begin{cases} \widetilde{z}^{t+1} &= z^t + (z^t - z^{t-1}) \\ z^{t+1} &= z^t + S\mathsf{Q}(\widetilde{z}^{t+1}) \end{cases} \quad . \qquad \text{(OFP Dual)}$$

**Energy function.** Using the new notation and dual variables $\{z^t\}$, we define an energy function $\psi : \mathbb{R}^2 \to \mathbb{R}$ over the new dual space $\mathbb{R}^2$: let $\psi((0,0)) = 0$, and for all other $z \in \mathbb{R}^2$, let

$$\psi(z) = \begin{cases} -\rho_1 \cdot z_1 + z_2 & \text{if } z \in \widehat{P}_2, & z_1 + z_2 & \text{if } z \in \widehat{P}_3, \\ -\rho_1 \cdot z_1 - \rho_2 \cdot z_2 & \text{if } z \in \widehat{P}_1, & z_1 - \rho_2 \cdot z_2 & \text{if } z \in \widehat{P}_4 \end{cases} \quad . \qquad (10)$$

It is also more expressive to write $\psi$ using the choice map $\mathsf{Q}$ from Definition 4.4. Recall for any $z \in \mathbb{R}^2$ that $\mathsf{Q}(z) \in \mathcal{W} = \{e_1, e_2, e_3, e_4\}$. Then we have the following equivalent definition of $\psi$:

**Definition 4.5.** For $z \in \mathbb{R}^2$, the function $\psi : \mathbb{R}^2 \to \mathbb{R}$ is given by

$$\psi(z) = \langle z, M\mathsf{Q}(z) \rangle \qquad \text{where} \quad M = \begin{pmatrix} -\rho_1 & -\rho_1 & 1 & 1 \\ -\rho_2 & 1 & 1 & -\rho_2 \end{pmatrix} \quad . \qquad (11)$$

Due to the relationship between the coordinates of $y_1^t$ and $y_2^t$ from Proposition 4.2, the following relationship between $\Psi$ and $\psi$ (and by Proposition 3.4, between $\Psi$ and $\text{Reg}(T)$) is immediate:

**Proposition 4.6.** *For all $t \geq 1$, $\Psi(y^t) = \psi(z^t)$. Moreover, $\text{Reg}(T) = \Psi(y^{T+1}) = \psi(z^{T+1})$.*

## 4.2 Bounded Energy of Subspace Dynamics

Leveraging Proposition 4.6, it suffices to derive an upper bound on $\psi(z^{T+1})$ in order to prove Theorem 3.5. For this, the key step is to prove a set of invariants on the dual iterates which establish that, if the magnitude of $\psi$ ever exceeds some constant threshold, the subsequent one-step change in $\psi$ is non-increasing (thus crystalizing the intuition from expression (5)).

For this, we first define an absolute constant $B$ as follows:

**Definition 4.7.** *Fix $A$ satisfying Assumption 1, and recall $a_{\max} = \|A\|_\infty$. Then define $B > 0$ by*

$$B = \min \left\{ b \in \mathbb{R}_+ : \text{ for all } z \in \mathbb{R}^2, \|z\|_1 \leq 6a_{\max} \implies \psi(z) \leq b \right\} .$$

In words, $B$ is the smallest constant whose sublevel set $\psi(z) \leq B$ contains an $\ell_1$ ball of radius $6a_{\max}$. Moreover, the magnitude of $B$ can be bounded from above as follows (see Section C for the proof):

**Proposition 4.8.** *Let $B$ be the constant from Definition 4.7. Then $B \leq 6a_{\max}(1 + \rho_1 + \rho_2)$.*

**Worst-case upper bound on energy.** Observe that if $\psi(z^t) \leq B$ for all times $t$, then the statement of Theorem 3.5 trivially holds. On the other hand, if the energy $\psi$ crosses this threshold under one step of the dynamics, then we have the following constant upper bound on $\psi(z^{t+1})$:

**Lemma 4.9.** *Suppose $\psi(z^t) \leq B$, and let $B' = 8a_{\max}(1 + \rho_1 + \rho_2)^2$. Then $\psi(z^{t+1}) \leq B'$*

**Cycling invariants and controlled energy growth.** The remaining step is to then control the energy growth whenever $\psi(z^t) > B$. For this, we prove the following key lemma:

**Lemma 4.10.** *Suppose $\psi(z^t) > B$ and $z^t \in \widehat{P}_i$ for $i \in [4]$. Then the following hold:*

*(1) Either (i) $\widetilde{z}^{t+1}, z^{t+1} \in \widehat{P}_i$ or (ii) $\widetilde{z}^{t+1}, z^{t+1} \in P_{i+1}$*

*(2) $\Delta\psi(z^t) = \psi(z^{t+1}) - \psi(z^t) \leq 0$.*

Part (1) of the lemma establishes invariants relating the true payoff vectors and predicted payoff vectors whenever energy is above the threshold $B$. Roughly speaking, when $\psi$ is larger than $B$, the dual vectors cycle consecutively through the regions $\widehat{P}_1, \ldots, \widehat{P}_4$ (similarly to the iterates of standard Fictitious Play), and this roughly implies that $\mathsf{Q}(\widetilde{z}^{t+1}) = \mathsf{Q}(z^{t+1})$.

Importantly, this alignment between $z^{t+1}$ and $\widetilde{z}^{t+1}$ is the key step needed to establish a non-increasing change in energy, as stated in part (2). To see why this is true, observe that using the definition of $\psi$ from (11), we can compute the one-step change $\Delta\psi(z^t) = \psi(z^{t+1}) - \psi(z^t)$ under (OFP Dual) as

$$\Delta\psi(z^t) = \big\langle z^{t+1}, M\mathbb{Q}(z^{t+1})\big\rangle - \big\langle z^t, M\mathbb{Q}(z^t)\big\rangle \tag{12}$$

$$= \big\langle z^t + S\mathbb{Q}(\widetilde{z}^{t+1}), M\mathbb{Q}(z^{t+1})\big\rangle - \big\langle z^t, M\mathbb{Q}(z^t)\big\rangle \tag{13}$$

$$= \underbrace{\big\langle z^t, M\big(\mathbb{Q}(z^{t+1}) - \mathbb{Q}(z^t)\big)\big\rangle}_{(a)} + \underbrace{\big\langle \mathbb{Q}(\widetilde{z}^{t+1}), S^\top M\mathbb{Q}(z^{t+1})\big\rangle}_{(b)} . \tag{14}$$

Here, (14) essentially encodes the expression for $\Delta\Psi$ from (5). In particular, straightforward calculations show that the matrix $S^\top M$ is skew-symmetric, and as Part (1) of the lemma roughly implies $\mathbb{Q}(z^{t+1}) = \mathbb{Q}(\widetilde{z}^{t+1})$, term (b) of (14) vanishes. Together with the column structure of $M$, the invariants of part (1) imply that part (a) of (14) is non-positive, and thus overall $\Delta\psi(z^t) \le 0$.

The full proofs of the preceding lemmas are developed in Section C and account more carefully for boundary conditions and tiebreaking. Figure 3 of Section C.3 also gives more visual intuition for the invariants and energy-growth behavior of Lemma 4.10. Granting these lemmas as true for now, we then give the proof of Theorem 3.5:

**Proof of Theorem 3.5.** Suppose for $t > 0$ that $\psi(z^{t-1}) \le B$ and $\psi(z^t) > B$. By Lemma 4.9, we must have $\psi(z^t) \le 8a_{\max}(1 + \rho_1 + \rho_2)^2$. Moreover, Lemmas C.2 and C.6 together imply that $\Delta\psi(z^t) \le 0$, and thus also $\psi(z^{t+1}) \le 8a_{\max}(1 + \rho_1 + \rho_2)^2$. It follows inductively that $\psi(z^{T+1}) \le 8a_{\max}(1 + \rho_1 + \rho_2)^2$. By definition, $\rho_1, \rho_2 \le (a_{\max}/a_{\mathrm{gap}})$, and thus we conclude

$$\psi(z^{T+1}) = \Psi(y^{T+1}) \le 8a_{\max}(1 + 2(a_{\max}/a_{\mathrm{gap}}))^2 . \qquad \square$$

## 5  Discussion and Conclusion

In this work, we established for the first time that the *unregularized* Optimistic Fictitious Play algorithm can obtain *constant* $O(1)$ regret in two-player zero-sum games. Our proof technique leverages a geometric viewpoint of Fictitious Play algorithms, and we believe the techniques established for the 2×2 regime can be extended to higher dimensions.

**Additional experimental results.**  To this end, in Table 2 we present additional experimental evidence indicating that constant regret bounds for Optimistic FP (similar to Theorem 3.1) hold more generally in higher-dimensional settings. The table shows the empirical regret of Optimistic FP and standard FP (using lexicographical tiebreaking) on three classes of zero-sum games, in three higher dimensional settings. For each setting, the algorithms were run from 100 random initializations, each for $T = 10000$ iterations, and we report the average regret over all initializations.

| *Dimension:* | **15×15** | | **25×25** | | **50×50** | |
|:---:|:---:|:---:|:---:|:---:|:---:|:---:|
| *Payoff Matrix* ↓ | **FP** | **OFP** | **FP** | **OFP** | **FP** | **OFP** |
| **Identity** | $155.1 \pm 3.9$ | $8.1 \pm 1.6$ | $161.3 \pm 3.1$ | $12.5 \pm 1.7$ | $167.2 \pm 2.5$ | $25.2 \pm 2.1$ |
| **RPS** | $235.6 \pm 7.6$ | $2.9 \pm 0.5$ | $242.2 \pm 6.3$ | $2.9 \pm 0.9$ | $242.7 \pm 5.9$ | $2.5 \pm 0.8$ |
| **Random [0, 1]** | $116.2 \pm 5.8$ | $4.3 \pm 0.8$ | $118.6 \pm 5.7$ | $5.7 \pm 0.9$ | $177.0 \pm 6.5$ | $13.0 \pm 1.5$ |

Table 2: Empirical regret of FP and OFP using lexicographical tiebreaking. Each entry reports an average and standard deviation (over 100 random initializations) of total regret after $T = 10000$ steps.

The results indicate that, in each class of payoff matrix and in each dimension, Opimistic FP has only constant regret compared to the regret of roughly $\sqrt{T} \approx 100$ obtained by standard FP. In Table 3 of Section E, we also report results using *randomized tiebreaking* for both algorithms and find similar conclusions, thus highlighting the robustness of the constant regret of OFP to tiebreaking rules. In Section E, we give more details on the experimental setup and additional plots similar to Figure 1.

**Limitations.** Formally proving whether Optimistic Fictitious Play obtains constant regret in all zero-sum games remains an important open question.

**Broader impact.** We acknowledge that there are many potential societal consequences of our theoretical results, however none of which we feel must be specifically highlighted.

**Acknowledgements.** JL and RS were supported by the MOE Tier 2 Grant (MOE-T2EP20223-0018), the National Research Foundation, Singapore, under its QEP2.0 programme (NRF2021-QEP2-02-P05), the National Research Foundation Singapore and DSO National Laboratories under the AI Singapore Programme (Award Number: AISG2-RP-2020-016), and SS was supported by the Villum Young Investigator Award (Grant no. 72091). The authors thank Anas Barakat and Andre Wibisono for helpful discussions.

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

## Table of Contents

# A    Details on Regret, Energy, and Fictitious Play

## A.1    Zero-Sum Games and Convergence to Nash Equilibrium

**Proposition 2.1.** *Let $\widetilde{x}_1^T = \frac{1}{T}(\sum_{t=0}^T x_1^t)$ and $\widetilde{x}_2^T = \frac{1}{T}(\sum_{t=0}^T x_2^t)$ denote the time-average iterates of Players 1 and 2, respectively, and suppose $\mathrm{Reg}(T) = o(T)$. Then $(\widetilde{x}_1^T, \widetilde{x}_2^T)$ converges (in duality-gap) to an NE of $A$ at a rate of $\mathrm{Reg}(T)/T = o(1)$.*

*Proof.* By definition of $\mathrm{Reg}(T)$ from (1), we have that

$$\frac{\mathrm{Reg}(T)}{T} \;=\; \max_{x \in \Delta_m} \; \Big\langle x, A\Big(\frac{1}{T} \sum_{t=0}^T x_2^t\Big)\Big\rangle - \min_{x \in \Delta_n} \; \Big\langle x, A^\top \Big(\frac{1}{T} \sum_{t=0}^T x_1^t\Big)\Big\rangle \;=\; \mathrm{DG}(\widetilde{x}_1^T, \widetilde{x}_2^T) \,,$$

where we use the definitions of $\widetilde{x}_1^T$ and $\widetilde{x}_2^t$, and of the duality gap $\mathrm{DG}(\cdot, \cdot)$. Thus if $\mathrm{Reg}(T) = o(\sqrt{T})$, then $\mathrm{DG}(\widetilde{x}_1^T, \widetilde{x}_2^T) = \frac{\mathrm{Reg}(T)}{T} = o(1)$.                            $\square$

## A.2    Proof of Proposition 3.4

In this section, we prove Proposition 3.4, which shows the equivalence between energy and regret. Restated here:

**Proposition 3.4.** *Let $\{x^t\}$ and $\{y^t\}$ be the iterates of (OFP). Then $\mathrm{Reg}(T) = \Psi(y^{T+1})$.*

*Proof.* For convenience, we let $d = m + n$, and we write $\mathcal{X} = \Delta_m \times \Delta_n$. Then recall from Definition 3.3 that for all $y \in \mathbb{R}^d$, the energy function $\Psi : \mathbb{R}^d \to \mathbb{R}$ is given by

$$\Psi(y) \;=\; \max_{x = (x_1, x_2) \in \mathcal{X}} \; \langle x, y \rangle \tag{15}$$

for $y = (y_1, y_2) \in \mathbb{R}^d$. Using the definitions of regret from from (1) and of the dual variables from Definition 2.3, we have

$$\begin{aligned}
\mathrm{Reg}(T) &= \max_{x_1 \in \Delta_m} \; \Big\langle x_1, \sum_{t=1}^T A x_2^t \Big\rangle - \min_{x_1 \in \Delta_n} \; \Big\langle x_2, \sum_{t=1}^T A^\top x_1^t \Big\rangle \\
&= \max_{x_1 \in \Delta_m} \; \big\langle x_1, y_1^{T+1} \big\rangle + \max_{x_2 \in \Delta_n} \; \big\langle x_2, y_2^{T+1} \big\rangle \\
&= \max_{x = (x_1, x_2) \in \mathcal{X}} \; \big\langle x, y^{T+1} \big\rangle \\
&= \Psi\big(y^{T+1}\big) \,.
\end{aligned}$$
                                               $\square$

## A.3    Details on Fictitious Play Variants as Skew-Gradient Descent

In this section, we give more details on the geometric viewpoint of Optimistic FP and standard FP introduced in Section 3.

**Dual dynamics of fictitious play variants.**    First, we recall that for a convex function $H : \mathbb{R}^d \to \mathbb{R}$ that its subgradient set at $y \in \mathbb{R}^d$ is defined as

$$\partial H(y) \;=\; \big\{ g \in \mathbb{R}^d : \forall z \in \mathbb{R}^d, H(z) \geq H(y) + \langle g, z - y \rangle \big\} \,. \tag{16}$$

Let $d = m + n$. Then for the energy function $\Psi$ from Definition 3.3, it follows that, for any $y \in \mathbb{R}^d$, the subgradient set $\partial \Psi(y)$ is the set of maximizers $\partial \Psi(y) = \mathrm{argmax}_{x \in \Delta_m \times \Delta_n} \langle x, y \rangle$. The next proposition (originally stated in Section 3) then follows by (1) using the definition of standard ($\alpha = 0$) and Optimistic FP ($\alpha = 1$) from ($\alpha$-OFP), and (2) by the definition of the dual payoff vectors.

**Proposition 3.6.** *Let $\{y^t\}$ denote the dual iterates of either standard Fictitious Play (e.g., ($\alpha$-OFP) with $\alpha = 0$) or Optimistic Fictitious Play. Then for all $t \geq 1$, the iterates of each algorithm evolve as*

$$\begin{cases} y^t \;=\; y^{t-1} + J x^{t-1} \\ x^t \;\in\; \partial \Psi\big(\widetilde{y}^{t+1}\big) \end{cases} \quad \text{where } J = \begin{pmatrix} 0 & A \\ -A^\top & 0 \end{pmatrix} \text{ and } \widetilde{y}^{t+1} = \begin{cases} y^t & \text{for FP} \\ 2y^t - y^{t-1} & \text{for OFP} \end{cases} , \tag{3}$$

*and it follows inductively that $y^{t+1} = y^t + J \partial \Psi(\widetilde{y}^{t+1})$, where $\partial \Psi(\widetilde{y}^{t+1})$ denotes a fixed vector in the subgradient set of $\Psi$ at $\widetilde{y}^{t+1}$.*

**One-step energy growth of FP variants.** Using Proposition 3.6, we can then derive the bounds on the one-step energy growth under FP and Optimistic FP, as stated in expressions (4) and (5).

For standard FP, using the convexity of $\Psi$ and Jensen's inequality (equivalently, the subgradient definition from (16)), and letting $\partial\Psi(y)$ denote a fixed vector in the subgradient set of $\Psi$ at $y$, we have for all $t \geq 1$

$$\Delta\Psi(y^t) = \Psi(y^{t+1}) - \Psi(y^t) \geq \langle\partial\Psi(y^t), y^{t+1} - y^t\rangle$$
$$= \langle\partial\Psi(y^t), J\partial\Psi(\widetilde{y}^{t+1})\rangle = \langle\partial\Psi(y^t), J\partial\Psi(y^t)\rangle = 0 .$$

Here, the first two equalities follow by the inductive update rule for FP from Proposition 3.6, and the final equality follows by skew-symmetry of $J = -J^\top$ (since $\langle y, Jy\rangle = 0$ for all $y \in \mathbb{R}^d$).

For Optimistic FP, again using the subgradient definition of expression (16), we have for $t \geq 1$:

$$\Delta\Psi(y^t) = \Psi(y^{t+1}) - \Psi(y^t) \leq \langle\partial\Psi(y^{t+1}), y^{t+1} - y^t\rangle$$
$$= \langle\partial\Psi(y^{t+1}), J\partial\Psi(\widetilde{y}^{t+1})\rangle ,$$

where the equality uses the update rule from Proposition 3.6 for Optimistic FP. Thus, by the skew-symmetry of $J$, and as explained in Section 3, the energy growth $\Delta\Psi(y^t)$ for Optimistic FP is non-positive whenever

$$\partial\Psi(y^{t+1}) = \partial\Psi(\widetilde{y}^{t+1}) = \partial\Psi(2y^t - y^{t-1}) .$$

# B  Assumptions on Payoff Matrix

Recall from Section 4 that in the proof of Theorem 3.5 for the 2×2 setting, we make the following assumption on the entries of the payoff matrix:

**Assumption 1.** Let $A \in \mathbb{R}^{2\times 2}$. Assume that

$$A = \begin{pmatrix} a & b \\ c & d \end{pmatrix} \quad \text{where} \quad \begin{cases} \text{(i)} & \det A = ad - bc = 0 \\ \text{(ii)} & a, d > \max\{0, b, c\} \end{cases} .$$

In this section, we give the proofs of Proposition 4.1 and Proposition 4.2. Proposition 4.1 establishes that the conditions of Assumption 1 hold without loss of generality, and Proposition 4.2 derives the resulting subspace invariance property of the payoff vectors under $A$.

## B.1  Proof of Proposition 4.1

We restate the proposition here:

**Proposition 4.1** (Bailey and Piliouras (2019))**.** *Let $A \in \mathbb{R}^{2\times 2}$ have a unique, interior NE, and let $\{x^t\}$ be the iterates of (OFP) on $A$. Then there exists $\widetilde{A} \in \mathbb{R}^{2\times 2}$ satisfying Assumption 1 such that (1) $\widetilde{A}$ and $A$ have the same NE and (2) the iterates $\{\widetilde{x}^t\}$ of running (OFP) on $\widetilde{A}$ are identical to $\{x^t\}$.*

The proof of Proposition 4.1 follows from the arguments in Bailey and Piliouras (2019, Appendix D). For completeness, we re-derive the full proof here.

*Proof.* Fix $A$, and let $(x_1^*, x_2^*) \in \Delta_2 \times \Delta_2$ denote its unique, interior equilibrium. Then the coordinates of $x_1^*$ and $x_2^*$ are given by

$$x_1^* = \left(\frac{d-c}{a+d-(b+c)}, \frac{a-b}{a+d-(b+c)}\right) \quad x_2^* = \left(\frac{d-b}{a+d-(b+c)}, \frac{a-c}{a+d-(b+c)}\right) . \quad (17)$$

Suppose that $A$ does not satisfy the conditions of Assumption 1. We will then construct $\widetilde{A} \in \mathbb{R}^{2\times 2}$ that both satisfies the assumption, and such that the two claims of the proposition statement hold.

For this, suppose that the entries of $A$ are shifted by the same additive constant $c$, and define the best responses $v$ and $v'$

$$v := \operatorname*{argmax}_{e_i \in \{e_1, e_2\}} \langle e_i, Ax\rangle \quad (18)$$

$$v' := \operatorname*{argmax}_{e_i \in \{e_1, e_2\}} \langle e_i, (A + c\mathbf{1})x\rangle = \operatorname*{argmax}_{e_i \in \{e_1, e_2\}} \langle e_i, Ax\rangle + c , \quad (19)$$

where $x \in \Delta_2$ and $\mathbf{1} \in \mathbb{R}^{2 \times 2}$ is the matrix of all ones. Thus for a fixed sequence of tiebreaking rules (e.g., the same adversarially-chosen tiebreak direction that is applied to determine $v$ is also applied to determine $v'$), it follows that the primal iterates of running (OFP) on $A$ will be identical to those of running (OFP) on $(A + c\mathbf{1})$ (and note the same argument holds for any FTRL algorithm or variant, including standard Fictitious Play and Alternating Fictitious Play).

Now suppose $\det A \neq 0$. Let $\widetilde{A}$ be the matrix

$$\widetilde{A} = A - \Big( \frac{\det A}{a + d - (b + c)} \Big) \cdot \mathbf{1} \ .$$

By straightforward calculations, it follows that $\det \widetilde{A} = 0$. Moreover, by (17), $\widetilde{A}$ has the same unique interior Nash equilibrium as $A$, and by the arguments above, the iterates of running (OFP) on $\widetilde{A}$ are equivalent to those on $A$. Thus without loss of generality, we assume $\det A = 0$.

We now establish that we can assume $a > \max\{0, b, c\}$ without loss of generality. First, we show $a \neq 0$ holds: by the assumption that $\det A = ad - bc = 0$, if $a = 0$, then $bc = 0$. However, by (17) and the assumption that $(x_1^*, x_2^*)$ is interior, we must have $a - c \neq 0 \implies c \neq 0$, which implies $b = 0$. This violates the constraint from (17) that $a - b = b \neq 0$, and thus without loss of generality, $a \neq 0$. To show without loss of generality that also $a > 0$, observe that the bilinear objective of the zero-sum game is given by

$$\max_{x_1 \in \Delta_2} \min_{x_2 \in \Delta_2} \langle x_1, A x_2 \rangle = \max_{x_1 \in \Delta_2} \min_{x_2 \in \Delta_2} -\langle x_1, -A x_2 \rangle = - \max_{x_2 \in \Delta_2} \min_{x_1 \in \Delta_2} \langle x_2, -A^\top x_1 \rangle \ .$$

Thus, by switching the maximization or minimization role between the players (via scaling the matrix by -1), we may assume that $a > 0$. Finally, to show $a > \max\{b, c\}$ holds without loss of generality, observe from (17) that if $a + d - (b + c) > 0$, then the interior Nash condition in (17) implies $a > c$ and $a > b$. If instead $a + d - (b + c) < 0$, then $0 < a < \min\{b, c\}$, and we can then rewrite the bilinear objective of the zero-sum game using a new payoff matrix with relabeled strategies (i.e., permuting the columns of $A$), as

$$\max_{x_1 \in \Delta_2} \min_{x_2 \in \Delta_2} \langle x_1, A x_2 \rangle = \max_{x_1 \in \Delta_2} \min_{x_2 \in \Delta_2} \langle x_1, A' x_2 \rangle \quad \text{where } A' = \begin{pmatrix} b & a \\ d & c \end{pmatrix} \ . \tag{20}$$

Under $A'$, we have $b + c - (a + d) > 0$, which from (17) and the reasoning above implies $b > \max\{a, d\} > 0$. As a consequence, by possibly permuting the columns of $A$ and relabeling the strategies of Player 1, we can assume in either case that $a > \max\{b, c\}$. Together, we conclude that the assumption $a > \max\{0, b, c\}$ holds without loss of generality.

Similarly, it follows that we may also assume $d > \max\{0, b, c\}$ without loss of generality. Specifically, using the relabeling argument above, we may assume $a + d - (b + c) > 0$. Then under the unique interior Nash and $\det A = 0$ assumptions, it follows from (17) (using similar arguments as for $a \neq 0$) that $d \neq 0$ and $d > \max\{b, c\}$. Moreover, as $a > 0$, if also $d < 0$, then this implies $c, b < 0$, meaning $\det A = ab - cd < 0$, contradicting the assumption that $\det A = 0$. Thus also $d > 0$, and we conclude that the assumption $d > \max\{0, b, c\}$ holds without loss of generality. $\qquad \square$

## B.2 Proof of Proposition 4.2

We restate the proposition here for convenience:

**Proposition 4.2.** *Let $A$ satisfy Assumption 1, and let $\{y_1^t\}$ and $\{y_2^t\}$ be the dual payoff vectors of (OFP). Then for every $t \geq 1$, it holds that $y_{12}^t = -\rho_1 \cdot y_{11}^t$ and $y_{22}^t = -\rho_2 \cdot y_{21}^t$, where $\rho_1 := (d - c)/(a - b) > 0$ and $\rho_2 := (d - b)/(a - c) > 0$.*

*Proof.* For player 1, let $v_1 = (d - c, a - b)$. Then observe that

$$A^\top v_1 = \begin{pmatrix} a & c \\ b & d \end{pmatrix} \begin{pmatrix} d - c \\ a - b \end{pmatrix} = \begin{pmatrix} ad - ac + ac - bc \\ bd - bc + ad - bd \end{pmatrix} = \begin{pmatrix} 0 \\ 0 \end{pmatrix} \ ,$$

where the final equality follows from the assumption that $\det A = ab - cd = 0$. Then for any $x \in \Delta_2$, we have

$$0 = \langle x, A^\top v_1 \rangle = \langle v_1, A x \rangle.$$

As $y_1^t = \sum_{k=1}^{t-1} A x_2^k$, this implies

$$\langle v_1, y_1^t \rangle \;=\; \sum_{k=1}^{t-1} \langle v_1, A x_2^k \rangle \;=\; 0 \;.$$

Thus for all $t$, we have $\langle v_1, y_1^t \rangle = (d-c) \cdot y_{11}^t + (a-b) \cdot y_{12}^t = 0$. Rearranging, and recalling that $\rho_1 := (d-c)/(a-b) > 0$ (where the inequality follows by Assumption 1), we find $y_{12}^t = -\rho_1 \cdot y_{11}^t$.

For the second player, let $v_2 = (d-b, a-c)$. Using a similar argument and calculation, we have $A v_2 = 0 \in \mathbb{R}^2$ and thus

$$\langle v_2, y_2^t \rangle \;=\; \sum_{k=1}^{t-1} \langle v_2, -A^\top x_1^k \rangle \;=\; \sum_{k=1}^{t-1} \langle x_1^k, -A v_2 \rangle \;=\; 0 \;.$$

For all $t$, it then follows that $\langle v_2, y_2^t \rangle = (d-b) \cdot y_{21}^t + (a-c) \cdot y_{22}^t = 0$, meaning $y_{22}^t = -\rho_2 \cdot y_{21}^t$. $\qquad\square$

## C  Proofs for Optimistic Fictitious Play Regret Upper Bound

In this section, we develop the omitted proofs from Section 4 that are needed to establish the main technical result of Theorem 3.5 (showing Optimistic FP has bounded energy in $2 \times 2$ games).

### C.1  Properties of the Energy Threshold $B$

In this section, we prove several properties related to the threshold $B$ that is used in the proof of Theorem 3.5: Recall that $B$ is defined as follows:

**Definition 4.7.** Fix $A$ satisfying Assumption 1, and recall $a_{\max} = \|A\|_\infty$. Then define $B > 0$ by

$$B = \min \left\{ b \in \mathbb{R}_+ : \text{ for all } z \in \mathbb{R}^2, \|z\|_1 \le 6 a_{\max} \implies \psi(z) \le b \right\} \;.$$

First, we prove the following upper bound on the magnitude of $B$ with respect to the constants $a_{\max}, \rho_1$, and $\rho_2$:

**Proposition 4.8.** *Let $B$ be the constant from Definition 4.7. Then $B \le 6 a_{\max}(1 + \rho_1 + \rho_2)$.*

*Proof.* By definition of $B$, the level set $\mathcal{L} = \{z \in R^2 : \psi(z) = B\}$ must intersect the boundary of the ball $\mathcal{B} = \{z \in \mathbb{R}^2 : \|z\|_1 \le B\}$ on at least one of the boundaries $P_{i \sim (i+1)}$ in the set $\widetilde{\mathcal{P}}$ from Definition 4.3. Let $\mathcal{I} = \widetilde{\mathcal{P}} \cap \mathcal{B} \cap \mathcal{L}$ be the intersection of these three sets. Using the definition of $\psi$ from (10) it follows that for $z \in \mathcal{I}$

$$B \;=\; \psi(z) \;=\; \begin{cases} \rho_1 \cdot |z_1| = \rho_1 \cdot 6 a_{\max} & \text{if } z \in P_{1 \sim 2} \\ |z_2| = 6 a_{\max} & \text{if } z \in P_{2 \sim 3} \\ |z_1| = 6 a_{\max} & \text{if } z \in P_{3 \sim 4} \\ \rho_2 \cdot |z_2| = \rho_2 \cdot 6 a_{\max} & \text{if } z \in P_{4 \sim 1} \end{cases} \;,$$

where in each case the equality comes from the fact that if $z \in \mathcal{I}$ then $\|z\|_1 = 6 a_{\max}$. It follows that

$$B \;\le\; 6 a_{\max} \cdot \max \left\{ 1, \rho_1, \rho_2 \right\} \;\le\; 6 a_{\max}(1 + \rho_1 + \rho_2) \;,$$

where the final inequality comes from the fact that $\rho_1, \rho_2 > 0$. $\qquad\square$

Next, we establish the following invariant:

**Proposition C.1.** *Let $B$ be the constant from Definition 4.3. Suppose $\psi(z) > B$ and suppose $z \in \widehat{P}_i \cup P_{(i-1)\,i}$ for some $i \in [4]$. Assume either $\widetilde{z} = z + S_j$ or $\widetilde{z} = z + S_j + S_k$ for $j, k \in [4]$ and $S$ as in (OFP Dual). Then*

$$\widetilde{z} \notin \widehat{P}_{i+2} \cup P_{(i+1)\sim(i+2)} \;. \tag{21}$$

*Proof.* We prove the claim for the case that $\widetilde{z} = z + S_j + S_k$, which by the same argument implies the result when $\widetilde{z} = z + S_j$. Without loss of generality, assume $i = 1$. Under the assumptions of the proposition, we will show that if $z \in \widehat{P}_1 \cup P_{4\sim 1}$, then $\widetilde{z} \notin \widehat{P}_3 \cup P_{2\sim 3}$. For this, observe first by definition of $B$ that if $\psi(z) > B$ then $\|z\|_1 > 6a_{\max}$. By definition of the sets $\widehat{P}_1$ and $\widehat{P}_3 \cup P_{2\sim 3}$, this implies that

$$\min_{z' \in \widehat{P}_3 \cup P_{2\sim 3}} \|z - z'\|_2 \ \geq \ \|z\|_2 \ \geq \ \frac{1}{\sqrt{2}} \|z\|_1 \ > \ \frac{6a_{\max}}{\sqrt{2}} \geq 4a_{\max} \ . \tag{22}$$

On the other hand, by construction of $\widetilde{z}$, and using the fact that $\|S\|_2 \leq 2a_{\max}$, we have

$$\|z - \widetilde{z}\|_2 \ \leq \ \|S_j\|_2 + \|S_k\|_2 \ \leq \ 2(2a_{\max}) = 4a_{\max} \ . \tag{23}$$

Then combining expressions (22) and (23), we find

$$\|z - \widetilde{z}\|_2 \ < \ \min_{z' \in \widehat{P}_3 \cup P_{2\sim 3}} \|z - z'\|_2 \ ,$$

and thus $\widetilde{z} \notin \widehat{P}_3 \cup P_{2\sim 3}$. $\qquad\square$

## C.2 Energy Upper Bound: Proof of Lemma 4.9

This section gives the proof of Lemma 4.9, which derives an upper bound on the energy $\psi(z^{t+1})$ when $\psi(z^t) \leq B$. Restated here:

**Lemma 4.9.** *Suppose $\psi(z^t) \leq B$, and let $B' = 8a_{\max}(1 + \rho_1 + \rho_2)^2$. Then $\psi(z^{t+1}) \leq B'$*

*Proof.* Using the definition of $\psi$ from (10), observe that

$$\psi(z^{t+1}) \ \leq \ \max \big\{ \max(\rho_1, \rho_2) \cdot \|z^{t+1}\|_1, \|z^{t+1}\|_1 \big\} \ \leq \ (1 + \rho_1 + \rho_2) \cdot \|z^{t+1}\|_1 \ . \tag{24}$$

Now recall by definition of the constant $B$ that $\psi(z^t) \leq B \implies \|z^t\|_1 \leq B$. Then as $\|z^{t+1}\|_1 = \|z^t + S_j\|_1$ for some $j \in [4]$, we have that

$$\begin{aligned}
\|z^{t+1}\|_1 \ &\leq \ \|z^t\|_1 + 2a_{\max} \ \leq \ B + 2a_{\max} \\
&\leq \ 6a_{\max}(1 + \rho_1 + \rho_2) + 2a_{\max} \\
&\leq \ 8a_{\max}(1 + \rho_1 + \rho_2) \ . \tag{25}
\end{aligned}$$

Here, the penultimate inequality follows from the upper bound on $B$ from Proposition 4.8, and the final inequality follows from the positivity of $\rho_1, \rho_2$.

Combining expressions (24) and (25), we conclude that

$$\psi(z^{t+1}) \ \leq \ 8a_{\max}(1 + \rho_1 + \rho_2)^2 \ . \qquad\square$$

## C.3 Cycling Invariants and Non-Increasing Energy Growth: Proof of Lemma 4.10

In this section, we develop the proof of Lemma 4.10, restated here:

**Lemma 4.10.** *Suppose $\psi(z^t) > B$ and $z^t \in \widehat{P}_i$ for $i \in [4]$. Then the following hold:*

(1) *Either (i) $\widetilde{z}^{t+1}, z^{t+1} \in \widehat{P}_i$ or (ii) $\widetilde{z}^{t+1}, z^{t+1} \in P_{i+1}$*

(2) $\Delta\psi(z^t) = \psi(z^{t+1}) - \psi(z^t) \ \leq \ 0.$

We give the proof of Lemma 4.10 in two parts: first in Lemma C.2 (Section C.3.1), we prove the invariants from Part (1). Then, in Lemma C.6 (Section C.3.2), we prove the non-positive energy growth bounds from Part (2).

In Figure 3, we also give visual intuition for the two claims of Lemma 4.10. In the figure, each subfigure shows the dual space $\mathbb{R}^2$, and the green region denotes the sublevel set $\psi(z) \leq B$. The left subfigure illustrates that for $\psi(z^t) > B$, the vectors $z^{t+1}$ and $\widetilde{z}^{t+1}$ will lie in the same region of $\mathcal{P}$, and thus $\Delta\psi(z^t) \leq 0$ (the latter point is captured by the fact that $z^{t+1}$ lies within the sublevel set $\psi(z) \leq B$). In contrast, as illustrated in the right subfigure, when $\psi(z^t) \leq B$, then in general the invariants of Part (1) of the lemma may not hold, and $\Delta\psi(z^t)$ can be strictly positive.

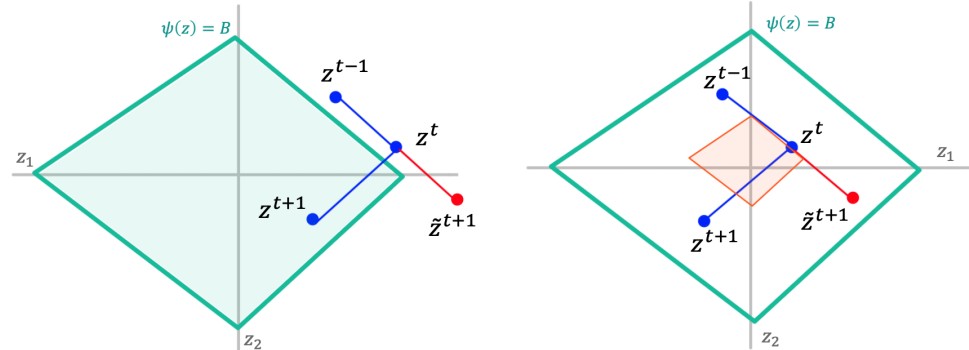

Figure 3: Visual intuition for the claims and proof of Lemma 4.10.

### C.3.1 Cycling Invariants: Part 1 of Lemma 4.10

In this section, we state and prove the following lemma, which establishes certain "cycling invariants" on the relationship between the predicted cost vector $\widetilde{z}^{t+1}$ and true cost vector $z^{t+1}$ that hold when the energy $\psi(z^t)$ is sufficiently large.

**Lemma C.2.** *Suppose $\psi(z^t) > B$ and $z^t \in \widehat{P}_i$ for $i \in [4]$. Then either*

(i) $\widetilde{z}^{t+1} \in \widehat{P}_i$ and $z^{t+1} \in \widehat{P}_i$, or

(ii) $\widetilde{z}^{t+1} \in P_{i+1}$ and $z^{t+1} \in P_{i+1}$.

**Proof of Lemma C.2.**   The proof of Lemma C.2 proceeds using three separate propositions, which we state and prove as follows:

First, we establish regions in which $\widetilde{z}^{t+1}$ and $z^{t+1}$ cannot lie when the energy $\psi$ is sufficiently large.

**Proposition C.3.** *Suppose $\psi(z^t) > B$ and $z^t \in \widehat{P}_i$ for some $i \in [4]$. Then*

$$\widetilde{z}^{t+1}, z^{t+1} \ \notin \ \widehat{P}_{i+2} \cup P_{(i+1)\sim(i+2)} \cup \widehat{P}_{i-1} \ .$$

*Proof.* We first show that $\widetilde{z}^{t+1}, z^{t+1} \notin \widehat{P}_{i+2} \cup P_{(i+1)\sim(i+2)}$. For this, recall that both $z^{t+1} = z^t + S_j$ and $\widetilde{z}^{t+1} = z^t + S_k$ for some $j, k \in [4]$. As $\psi(z^t) > B$, then applying Proposition C.1 implies that both $z^{t+1}, \widetilde{z}^{t+1} \notin \widehat{P}_{i+2} \cup P_{(i+1)\sim(i+2)}$.

Next, we establish that $z^{t+1} \notin \widehat{P}_{i-1}$. For this, as $z^t \in \widehat{P}_i$, then either (a) $\widetilde{z}^{t+1} \in \widehat{P}_{i-1}$ or (b) $\widetilde{z}^{t+1} \in \widehat{P}_{i+2} \cup P_{(i+1)\sim(i+2)}$. We have already established case (b) cannot hold. Similarly, as $\psi(z^t) > B$ implies $\|z^t\|_1 > 3a_{\max}$, then case (a) implies that either $\psi(z^t) \le B$ or $z^{t+1} \notin \widehat{P}_{i-1}$, which is a contradiction. So $z^{t+1} \notin \widehat{P}_{i-1}$.

Similarly, to have $\widetilde{z}^{t+1} \in \widehat{P}_{i-1}$, then by definition of $S$ we must have either (a) $z^t - z^{t-1} = S_{i+2}$ or (b) $z^t - z^{t-1} = S_{i+1}$. Case (a) implies $\widetilde{z}^t \in \widehat{P}_{i+2} \cup P_{(i+1)\sim(i+2)}$, which cannot hold due to Proposition C.1. Case (b) implies $\widetilde{z}^t \in \widehat{P}_{i+1} \cup P_{i\sim(i+1)}$, which again due to the definition of $B$ contradicts that either $\psi(z^t) > B$ or that $\widetilde{z}^{t+1} \in \widehat{P}_{i-1}$. Thus we conclude $\widetilde{z}^{t+1} \notin \widehat{P}_{i-1}$. $\qquad\square$

Proposition C.3 establishes that if $\psi(z^t) > B$, then we must have $\widetilde{z}^{t+1}, z^{t+1} \in \widehat{P}_i \cup P_{i+1}$. Thus to conclude the proof of Lemma C.2, it suffices to establish that $\widetilde{z}^{t+1} \in \widehat{P}_i \implies z^{t+1} \in \widehat{P}_i$ and that $\widetilde{z}^{t+1} \in P_{i+1} \implies z^{t+1} \in P_{i+1}$. We prove these claims in the following two propositions:

**Proposition C.4.** *Suppose $\psi(z^t) > B$ and $z^t \in \widehat{P}_i$ for some $i \in [4]$. Then:*

$$\widetilde{z}^{t+1} \in \widehat{P}_i \implies z^{t+1} \in \widehat{P}_i \ .$$

*Proof.* We distinguish the cases when $\widetilde{z}^{t+1} \in P_i$ and $\widetilde{z}^{t+1} \in P_{i\sim(i+1)}$. In the first case, if $\widetilde{z}^{t+1} \in P_i$, then by definition $z^{t+1} = z^t + S_i$. If $\psi(z^{t+1}) = \psi(z^t)$, then $z^{t+1} = \widetilde{z}^{t+1} \in P_i$. If instead $\psi(z^{t+1}) \neq \psi(z^t)$, then $\widetilde{z}^{t+1} = z^t - z^{t-1} = S_j$ for some $j \neq i \in [4]$. However, by definition of the constant $B$, we must have $z^{t+1} \notin P_{i+1}$, as otherwise the assumption $\psi(z^t) > B$ would be violated. Then Proposition C.3 implies $z^{t+1} \in \widehat{P}_i$. Thus if $\widetilde{z}^{t+1} \in P_i$, then $z^{t+1} \in \widehat{P}_i$.

For the second case, suppose $\widetilde{z}^{t+1} \in P_{i\sim(i+1)}$. Then we must have $\widetilde{z}^{t+1} - z^t \in \{S_i, S_{i+1}\}$. Moreover, recall that $Q(\widetilde{z}^{t+1}) \in \{e_i, e_{i+1}\}$. In either case, using the structure of adjacent columns $i$ and $i+1$ of $S$, it follows that $z^{t+1} \in P_{i\sim(i+1)} \subset \widehat{P}_i$. Thus if $\widetilde{z}^{t+1} \in P_{i\sim(i+1)}$, then also $z^{t+1} \in \widehat{P}_i$, which concludes the proof. $\qquad\square$

**Proposition C.5.** *Suppose $\psi(z^t) > B$ and $z^t \in \widehat{P}_i$ for some $i \in [4]$. Then:*

$$\widetilde{z}^{t+1} \in P_{i+1} \implies z^{t+1} \in P_{i+1} .$$

*Proof.* Suppose $z^t \in P_i$. If $\widetilde{z}^{t+1} \in P_{i+1}$, then $Q(z^{t+1}) = e_{i+1}$, and also $\widetilde{z}^{t+1} - z^t \in \{S_i, S_{i+1}\}$. Using the structure of adjacent columns $i$ and $i+1$ of $S$, it then follows that $z^{t+1} = z^t + S_{i+1} \in P_{i+1}$. Similarly, if instead $z^t \in P_{i\sim(i+1)}$, then $z^{t+1} = z^t + S_{i+1} \in P_{i+1}$ by definition of $S$. Thus in either case, if $\widetilde{z}^{t+1} \in P_{i+1}$, then also $z^{t+1} \in P_{i+1}$. $\qquad\square$

### C.3.2 Non-Increasing Energy Growth: Part (2) of Lemma 4.10

In this section we state and prove the following lemma, which gives non-positive bounds on the energy growth under the two cases in Part (1) of Lemma 4.10:

**Lemma C.6.** *Fix $i \in [4]$, and suppose $z^t \in \widehat{P}_i$. Suppose that either (i) $\widetilde{z}^{t+1} \in \widehat{P}_i$ and $z^{t+1} \in \widehat{P}_i$ or (ii) $\widetilde{z}^{t+1} \in P_{i+1}$ and $z^{t+1} \in P_{i+1}$. Then $\Delta\psi(z^t) \leq 0$.*

*Proof.* To start, we rederive the one-step change in energy growth under (OFP Dual).

**One-step change in energy:** Using (OFP Dual) and Definition 4.5, we have for any $t \geq 1$:

$$\Delta\psi(z^t) = \psi(z^{t+1}) - \psi(z^t) \tag{26}$$

$$= \langle z^{t+1}, MQ(z^{t+1}) \rangle - \langle z^t, MQ(z^t) \rangle \tag{27}$$

$$= \langle z^t + SQ(\widetilde{z}^{t+1}), MQ(z^{t+1}) \rangle - \langle z^t, MQ(z^t) \rangle \tag{28}$$

$$= \underbrace{\langle z^t, M(Q(z^{t+1}) - Q(z^t)) \rangle}_{\text{(a)}} + \underbrace{\langle Q(\widetilde{z}^{t+1}), S^\top MQ(z^{t+1}) \rangle}_{\text{(b)}} . \tag{29}$$

By the definitions of $S$ and $M$ from expressions (OFP Dual) and (11), respectively, recalling that $\rho_1 = (d-c)/(a-b)$ and $\rho_2 = (d-b)/(a-c)$, and using the fact from Assumption 1 that $\det A = ab - cd = 0$, we can compute

$$S^\top M = \begin{pmatrix} b & -c \\ a & -c \\ a & -a \\ b & -a \end{pmatrix} \begin{pmatrix} -\rho_1 & -\rho_1 & 1 & 1 \\ -\rho_2 & 1 & 1 & -\rho_2 \end{pmatrix} = \begin{pmatrix} 0 & d-c & b-c & b-d \\ c-d & 0 & a-c & a-d \\ b-c & c-a & 0 & a-b \\ d-b & d-a & b-a & 0 \end{pmatrix} . \tag{30}$$

Thus, expression (30) shows $S^\top M$ is skew-symmetric.

**Proof for Case (i):** To prove the claim for case (i) of the lemma, we start with the case that $z^t \in P_i$ and also $\widetilde{z}^{t+1}, z^{t+1} \in P_i$. Then by definition of $Q$, we have $Q(z^t) = Q(\widetilde{z}^{t+1}) = Q(\widetilde{z}^t) = e_i$. By skew-symmetry of $S^\top M$, observe in part (b) of expression (29) that

$$\langle Q(\widetilde{z}^{t+1}), S^\top MQ(z^{t+1}) \rangle = \langle Q(\widetilde{z}^{t+1}), S^\top MQ(z^{t+1}) \rangle = 0 .$$

Moreover, in part (a) of expression (29), we also have

$$\langle z^t, M(Q(z^{t+1}) - Q(z^t)) \rangle = \langle z^t, M(Q(z^t) - Q(z^t)) \rangle = 0 ,$$

and thus $\Delta\psi(z^t) = 0$. In the case that $z^t \in P_i$ and $\widetilde{z}^{t+1} \in P_{i\sim(i+1)}$, then observe from the structure of $S$ that we must also have $z^{t+1} \in P_{i\sim(i+1)}$. Then by definition of $\psi$, for any $z \in P_{i\sim(i+1)}$, we have $\psi(z) = \langle z, Me_i \rangle$. Thus we can rewrite expression (26) as

$$\Delta\psi(z^t) = \langle z^t + SQ(\widetilde{z}^{t+1}), Me_i \rangle - \langle z^t, MQ(z^t) \rangle \tag{31}$$

$$= \langle z^t, M(e_i - Q(z^t)) \rangle + \langle Q(\widetilde{z}^{t+1}), S^\top Me_i \rangle . \tag{32}$$

As $z^t \in P_i \implies Q(z^t) = e_i$, the first term above vanishes. Moreover, as $\widetilde{z}^{t+1} \in P_{i\sim(i+1)}$, we have $Q(\widetilde{z}^{t+1}) \in \{e_i, e_{i+1}\}$. By skew-symmetry of $S^\top M$, if $Q(\widetilde{z}^{t+1}) = e_i$, then the second term of (32) also vanishes. On the other hand, if $Q(\widetilde{z}^{t+1}) = e_{i+1}$, then the second term is negative, which follows from the fact that, under Assumption 1, each entry $(S^\top M)_{i+1,i} < 0$. In either case, we find $\Delta\psi(z^t) \leq 0$.

Finally, observe that if $z^t \in P_{i\sim(i+1)}$, then by definition of (OFP Dual), we cannot have both $\widetilde{z}^{t+1}, z^{t+1} \in \widehat{P}_i$. Thus the conditions of case (i) do not apply, which concludes the proof of the lemma under case (i).

**Proof for Case (ii)**: To prove the claim for case (ii), suppose first that $z^t \in P_i$ and thus $Q(z^t) = e_i$. By the assumptions of claim (ii), we also have $Q(\widetilde{z}^{t+1}) = Q(z^{t+1}) = e_{i+1}$. Thus it again follows by skew-symmetry of $S^\top M$ that for part (b) of expression (29)

$$\langle Q(\widetilde{z}^{t+1}), S^\top MQ(z^{t+1}) \rangle = 0 . \tag{33}$$

For part (a) of (29), by case analyis on the columns of $M$, it follows that when $Q(z^t) = e_i$ and $Q(z^{t+1}) = e_{i+1}$, then

$$\langle z^t, M(Q(z^{t+1}) - Q(z^t)) \rangle = \begin{cases} (1 + \rho_2) \cdot z_2^t & \text{if } Q(z^t) = e_1 \\ (1 + \rho_1) \cdot z_1^t & \text{if } Q(z^t) = e_2 \\ -(1 + \rho_2) \cdot z_2^t & \text{if } Q(z^t) = e_3 \\ -(1 + \rho_1) \cdot z_1^t & \text{if } Q(z^t) = e_4 \end{cases} . \tag{34}$$

Given the definition of Q, it follows that $Q(z^t) = e_1 \implies z_2^t \leq 0$, that $Q(z^t) = e_2 \implies z_1^t \leq 0$, that $Q(z^t) = e_3 \implies z_2^t \geq 0$, and that $Q(z^t) = e_4 \implies z_1^t \geq 0$. Together with the fact that $\rho_1, \rho_2 > 0$ by definition, in each case of expression (34), we find $\langle z^t, M(Q(z^{t+1}) - Q(z^t)) \rangle \leq 0$. Together with (33), this means $\Delta\psi(z^t) \leq 0$.

In the case that $z^t \in P_{i\sim(i+1)}$, then either $Q(z^t) = e_i$ or $Q(z^t) = e_{i+1}$. If the latter holds, given that also $Q(z^{t+1}) = e_{i+1}$ by assumption, then part (a) of (29) is trivially 0. If the former holds, we recover the cases of expression (34), and thus part (a) of (29) is non-positive. In either case, part (b) of (29) remains 0 as in expression (33), and thus we conclude that $\Delta\psi(z^t) \leq 0$. This proves the lemma under case (ii). $\square$

# D   Proofs for Alternating Fictitious Play Regret Lower Bound

In this section, we develop the proof of Theorem 3.2, which gives a *lower bound* of $\Omega(\sqrt{T})$ on the regret of *Alternating* Fictitious Play. Restated here:

**Theorem 3.2.** *Suppose $x_1^1 = (p, 1-p) \in \Delta_2$ for irrational $p \in (3/4, 1)$, and let $\{x^t\}$ be the iterates of Alternating FP on* (Matching Pennies) *using any tiebreaking rule. Then* $\mathrm{Reg}^{\mathrm{alt}}(T) \geq \Omega(\sqrt{T})$.

The organization of this section is as follows: in Section D.1 we recall the setup of alternating play in zero-sum games, as well as on the notion of alternating regret. In Section D.2, we formally define the Alternating Fictitious Play algorithm. Finally, in Section D.3, we give the proof of Theorem 3.2.

## D.1   Details on Alternating Play and Alternating Regret

**Alternating play.**   We consider the model of alternating online learning in two-player zero-sum games as in Bailey et al. (2020); Wibisono et al. (2022); Katona et al. (2024). Defined formally:

**Definition D.1** (Alternating Play). Fix a payoff matrix $A \in \mathbb{R}^{m \times n}$. Over $T$ rounds, Players 1 and 2 alternate updating their strategies $x_1^t \in \Delta_m$ and $x_2^t \in \Delta_n$ as follows:

- **(Initialization)** Assume without loss of generality $T$ is even. At time $t = 1$, Player 1 chooses an initial $x_1^1 \in \Delta_m$, and Player 2 observes $-A^\top x_1^1$.

- **(Even rounds – Player 2 updates)** When $t = 2k$ (for $k \geq 1$):

  Player 1 sets $x_1^t = x_1^{t-1} \in \Delta_m$      Player 2 updates $x_2^t \in \Delta_n$.

  Player 1 observes $Ax_2^t$              Player 2 observes $-A^\top x_1^{t-1}$.

- **(Odd rounds – Player 1 updates)** When $t = 2k + 1$ (for $k \geq 1$):

  Player 1 updates $x_1^t \in \Delta_m$       Player 2 sets $x_2^t = x_2^{t-1} \in \Delta_n$.

  Player 1 observes $Ax_2^{t-1}$         Player 2 observes $-A^\top x_1^t$.

**Alternating regret.** Under alternating play, we now measure the performance of each player by its *alternating regret* (Wibisono et al., 2022; Cevher et al., 2023; Hait et al., 2025). For this, first observe under alternating play that each player's *cumulative payoff* can be written as:

$$\text{Player 1 cumulative payoff:} \quad \sum_{k=1}^{T/2} \left\langle x_1^{2k-1}, A(x_2^{2k} + x_2^{2k-2}) \right\rangle .$$

$$\text{Player 2 cumulative payoff:} \quad \sum_{k=1}^{T/2} \left\langle x_2^{2k}, -A^\top(x_1^{2k+1} + x_1^{2k-1}) \right\rangle . \tag{35}$$

Here and throughout, we assume for notational convenience that $x_2^0 = 0 \in \mathbb{R}^n$ and $x_1^{T+1} = 0 \in \mathbb{R}^m$. Then alternating regret is defined as follows:

**Definition D.2** (Alternating Regret). Let $T$ be even. Define $\text{Reg}_1^{\text{alt}}(T)$ and $\text{Reg}_2^{\text{alt}}(T)$ as

$$\text{Reg}_1^{\text{alt}}(T) = \max_{x \in \Delta_m} \sum_{k=1}^{T/2} \left\langle x - x_1^{2k-1}, A(x_2^{2k} + x_2^{2k-2}) \right\rangle$$

$$\text{Reg}_2^{\text{alt}}(T) = \min_{x \in \Delta_n} \sum_{k=1}^{T/2} \left\langle x_2^{2k} - x, A^\top(x_1^{2k+1} + x_1^{2k-1}) \right\rangle .$$

Then define $\text{Reg}^{\text{alt}}(T) = \text{Reg}_1^{\text{alt}}(T) + \text{Reg}_2^{\text{alt}}(T)$.

Similar to standard (simultaneous) play, sublinear regret bounds for $\text{Reg}^{\text{alt}}(T)$ correspond to convergence of the time-average iterates under alternating play to a Nash equilibrium of $A$. For this, define the time-average iterates $\widetilde{x}_1^T \in \Delta_m$ and $\widetilde{x}_2^T \in \Delta_n$ by

$$\widetilde{x}_1^T = \frac{1}{T} \left( \sum_{k=1}^{T/2} x_1^{2k-1} + x_1^{2k+1} \right) \quad \text{and} \quad \widetilde{x}_2^T = \frac{1}{T} \left( \sum_{k=1}^{T/2} x_2^{2k-2} + x_2^{2k+2} \right) . \tag{36}$$

Then we have the following proposition (analogous to Proposition 2.1 for simultaneous play):

**Proposition D.3.** *Fix $A \in \mathbb{R}^{m \times n}$. Let $\widetilde{x}_1^T \in \Delta_m$ and $\widetilde{x}_2^T \in \Delta_n$ denote the time-average iterates under the alternating play of Definition D.2, as in expression (36). Suppose $\text{Reg}^{\text{alt}}(T) \leq \alpha = o(T)$. Then $(\widetilde{x}_1^T, \widetilde{x}_2^T)$ converges in duality gap to an NE of $A$ at a rate of $\alpha/T = o(1)$.*

*Proof.* By definition of the player-wise cumulative costs from (35) (and recalling that we set $x_2^0 = 0 \in \mathbb{R}^n$ and $x_1^{T+1} = 0 \in \mathbb{R}^m$ for notational convenience), observe that

$$\sum_{k=1}^{T/2} \left\langle x_1^{2k-1}, A(x_2^{2k} + x_2^{2k-2}) \right\rangle + \sum_{k=1}^{T/2} \left\langle x_2^{2k}, -A^\top(x_1^{2k+1} + x_1^{2k-1}) \right\rangle = 0 .$$

It follows from the Definition D.2 that

$$\text{Reg}^{\text{alt}}(T) = \text{Reg}_1^{\text{alt}}(T) + \text{Reg}_2^{\text{alt}}(T)$$

$$= \max_{x \in \Delta_m} \sum_{k=1}^{T/2} \left\langle x, A(x_2^{2k} + x_2^{2k-2}) \right\rangle - \min_{x \in \Delta_n} \sum_{k=1}^{T/2} \left\langle x, A^\top(x_1^{2k+1} + x_1^{2k-1}) \right\rangle$$

$$= \max_{x \in \Delta_m} \left\langle x, A(T \cdot \widetilde{x}_2^T) \right\rangle - \min_{x \in \Delta_n} \left\langle x, A^\top(T \cdot \widetilde{x}_1^T) \right\rangle \leq \alpha ,$$

where in the final line we use the definition of $\widetilde{x}_1^T$ and $\widetilde{x}_2^T$ from (36) and the assumption that $\text{Reg}(T) \leq \alpha$. Then dividing by $T$ gives

$$\text{DG}(\widetilde{x}_1^T, \widetilde{x}_2^T) \;=\; \max_{x \in \Delta_m} \; \left\langle x, A\widetilde{x}_2^T \right\rangle - \min_{x \in \Delta_n} \; \left\langle x, A^\top \widetilde{x}_1^T \right\rangle \;\leq\; \frac{\alpha}{T} \;,$$

which yields the statement of the proposition. $\qquad\qquad\qquad\qquad\qquad\qquad\qquad\square$

## D.2    Details on Alternating Fictitious Play

Under the alternating play setup of Definition D.1, we now specify the Alternating Fictitious Play algorithm. For any even $t \geq 2$, the primal iterates of Players 1 and 2 at times $t+1$ and $t+2$ update according to

$$x_1^{t+1} \;:=\; \underset{x \in \{e_i\}_m}{\text{argmax}} \; \left\langle x, \sum_{k=1}^{t/2} A(x_2^{2k} + x_2^{2k-2}) \right\rangle \quad \text{and} \quad x_2^{t+1} = x_2^t$$

$$x_2^{t+2} \;:=\; \underset{x \in \{e_i\}_n}{\text{argmax}} \; \left\langle x, \sum_{k=1}^{t/2} -A^\top(x_1^{2k+1} + x_2^{2k-1}) \right\rangle \quad \text{and} \quad x_1^{t+2} = x_1^{t+1} \;.$$

In other words, as in standard Fictitious Play (c.f., ($\alpha$-OFP) for $\alpha = 0$), in Alternating Fictitious Play each player (in an alternating fashion), selects the best-response to the cumulative observed payoff vectors over all prior rounds.

**Primal-Dual update for Alternating FP.**    Similar to the analysis for Optimistic FP, define the dual payoff vectors $y_1^t = \sum_{k=1}^{t-1} A x_2^k \in \mathbb{R}^m$ and $y_2^t = \sum_{k=1}^{t-1} -A^\top x_1^k \in \mathbb{R}^n$. Then the iterates of Alternating FP can be equivalently expressed as follows:

**Definition D.4.**    Assume the alternating play setting of Definition D.1. Let $y_1^2 = 0 \in \mathbb{R}^m$, and let $y_2^2 = -A^\top x_1^1 \in \mathbb{R}^n$. Then for $t \geq 2$, the dual (i.e., $(y_1^t, y_2^t)$) and primal (i.e., $(x_1^t, x_2^t)$) iterates of Alternating FP are given by

$$
\begin{array}{llll}
(t \text{ even}) & \begin{cases} x_1^t = x_1^{t-1} \\ x_2^t = \text{argmax}_{x \in \{e_i\}_n} \left\langle x, y_2^t \right\rangle \end{cases} & \text{and} & \begin{cases} y_1^{t+1} = y_1^t + A x_2^t \\ y_2^{t+1} = y_2^t - A^\top x_1^{t-1} \;. \end{cases} \\[18pt]
(t \text{ odd}) & \begin{cases} x_1^t = \text{argmax}_{x \in \{e_i\}_m} \left\langle x, y_1^t \right\rangle \\ x_2^t = x_2^{t-1} \end{cases} & \text{and} & \begin{cases} y_1^{t+1} = y_1^t + A x_2^{t-1} \\ y_2^{t+1} = y_2^t - A^\top x_1^t \;. \end{cases}
\end{array}
\tag{AFP}
$$

Moreover, recall the energy function $\Psi$ from Definition 3.3 and $\text{Reg}^{\text{alt}}(T)$ from Definition D.2. Then, analogously to Proposition 3.4, following equivalence between energy and alternating regret holds:

**Proposition D.5.**    *Let $\{x^t\}$ and $\{y^t\}$ be iterates of (AFP). Then $\text{Reg}^{\text{alt}}(T) = \Psi(y^{T+1})$.*

## D.3    Proof of Theorem 3.2: Regret Lower Bound on Matching Pennies

We now prove the lower bound on the regret of (AFP) on Matching Pennies. For this, recall that the Matching Pennies payoff matrix is given by

$$A \;=\; \begin{pmatrix} 1 & -1 \\ -1 & 1 \end{pmatrix} \;. \qquad\qquad\qquad \text{(Matching Pennies)}$$

**Subspace Dynamics of AFP for Matching Pennies.**    It is straightforward to check that (Matching Pennies) satisfies the conditions of Assumption 1. Moreover, this also implies Proposition 4.2 holds for the dual iterates of (AFP), in particular for $\rho_1 = \rho_2 = 1$.

Thus, to prove the theorem, we reuse the components of the *subspace dynamics* introduced in Section 4. Specfically, we reuse the notation of the primal and dual iterates $\{w^t\}$ and $\{z^t\}$, as well as the choice map Q from Definition 4.4, and the energy $\psi$ from Definition 4.5.

Under (Matching Pennies), it is then straightforward to check that the matrix $S$ from (9) and the energy $\psi$ from Definition 4.5 are given by:

$$S = \begin{pmatrix} -1 & 1 & 1 & -1 \\ 1 & 1 & -1 & -1 \end{pmatrix} \quad \text{and for all } z \in \mathbb{R}^2 \colon \psi(z) = \|z\|_1 \;.$$

Similarly to Proposition 4.6, and using the definition of $\psi$ and the iterates $\{z^t\}$, we also have the following relationship between $\Psi$ and $\psi$:

**Proposition D.6.** *Let $\{y^t\}$ be the iterates of* (AFP) *on* (Matching Pennies)*, and let $\{z^t\}$ be the corresponding subspace iterates. Then $\Psi(y^{T+1}) = \psi(z^{t+1})$.*

Moreover, under the primal-dual definition of (9) it follows inductively (and using the definition of $\{w^t\}$, $\{z^t\}$, and Q) that for all $t \geq 3$:

$$w^t \;=\; \begin{cases} \mathrm{Q}((z_1^{t-1}, z_2^t)) & \text{for } t \text{ even} \\ \mathrm{Q}((z_1^t, z_2^{t-1})) & \text{for } t \text{ odd} \end{cases} . \tag{37}$$

Then for $t \geq 3$ that the dual iterates $\{z^t\}$ can be further rewritten as

$$z^{t+1} \;=\; z^t + S\mathrm{Q}(\widetilde{z}^{t+1}) \quad \text{where } \widetilde{z}^{t+1} \;=\; \begin{cases} (z_1^{t-1}, z_2^t) & \text{for } t \text{ even} \\ (z_1^t, z_2^{t-1}) & \text{for } t \text{ odd} \end{cases} . \tag{AFP Dual}$$

Thus, similar to (AFP Dual), the subspace iterates of Alterating Fictitious Play can be expressed with respect to a predicted payoff vector $\widetilde{z}^{t+1}$. Now, due to the alternating play setting, the position of this predicted vector depends on the parity of $t$.

**Overall proof strategy.** Given the equivalence between $\mathrm{Reg}^{\mathrm{alt}}(T)$ and $\Psi(y^{T+1})$ from Proposition D.5, and on the equivalence between $\Psi(y^{T+1})$ and $\psi(z^{T+1})$ from D.6, to prove Theorem 3.2, it suffices to establish the following lower bound on the energy $\psi(z^{T+1})$:

**Lemma D.7.** *Assume the setting of Theorem 3.2, and let $\{z^t\}$ be the dual iterates of* (AFP Dual)*. Then $\psi(z^{T+1}) \geq \Omega(\sqrt{T})$.*

To prove Lemma D.7, we introduce a *phase structure* (in similar spirit to the analysis of Lazarsfeld et al. (2025)), where each phase tracks a subsequence of consecutive time steps where the iterates $\{w^t\}$ are at the same primal vertex. Formally, we define:

**Definition D.8.** *Let $\{w^t\}$ be the primal iterates from (37), and fix $t_0 = 2$. For $k \geq 1$, let $t_k := \min\{t > t_{k-1} : w^t \neq w^{t_{k-1}}\}$. Then define Phase $k$ as the subsequence of iterates from times $t = t_k, t_k + 1 \ldots, t_{k+1} - 1$, and let $\tau_k = t_{k+1} - t_k$ be the length of the phase. Let $K \geq 0$ denote the total number of phases in $T$ rounds such that $T = \sum_{k=0}^{T} \tau_k$.*

Using the phase setup of Definition D.8, the core technical component of proving Lemma D.7 is to establish the following proposition:

**Proposition D.9.** *Assume the setting of Theorem 3.2. Then for each Phase $k = 1, \ldots, K$, the following hold:*

    *(i) $\psi(z^{t_k}) \leq \psi(z^{t_{k-1}}) + 2$*

    *(ii) $\tau_k = \Theta(\psi(z^{t_k}))$.*

*Moreover, for at least $K/2$ phases $k$, it holds that (iii) $\psi(z^{t_k}) \geq \psi(z^{t_{k-1}}) + 1$.*

The proof of Proposition D.9 is developed in Section D.3.1. Granting the claims of the proposition as true for now, we give the proof of Lemma D.7 (and thus also of Theorem 3.2):

*Proof (of Lemma D.7).* By claim (iii) of Proposition D.9, the energy $\psi$ is strictly increasing in at least $K/2$ phases, and thus

$$\psi(z^{T+1}) \;\geq\; \frac{K}{2} . \tag{38}$$

To prove the statement of the lemma, it then suffices to derive a lower bound on $K$. For this, recall by Definition D.8 that $T = \sum_{k=1}^{K} \tau_k$. Moreover, combining claims (i) and (ii) of Proposition D.9, we find for all $k$ that $\tau_k = \Theta(\psi(z^{t_k})) \leq \Theta(\psi(z^{t_{k-1}}) + 2) \leq \Theta(k)$. Combining these pieces, we have

$$T \;=\; \sum_{k=1}^{K} \tau_k \;\leq\; \sum_{k=1}^{K} \Theta(k) \;\leq\; \Theta(K^2) . \tag{39}$$

Thus $K^2 \geq \Omega(T) \implies K \geq \Omega(\sqrt{T})$. Substituting into (38), we conclude $\psi(z^{T+1}) \geq \Omega(\sqrt{T})$. $\square$

### D.3.1 Proof of Proposition D.9

We now prove the claims of Proposition D.9. For this, we start by establishing the following invariant between the dual iterates $z^{t-1}, z^t, z^{t+1}$ and the predicted vector $\widetilde{z}^{t+1}$.

**Analysis of initial phases.** We begin by computing the dual iterates during the first two phases, which helps to both give intuition for the energy growth behavior of Alternating FP, as well as to streamline the remainder of the proof. For this, recall that initially $x_2^1 = (p, 1-p) \in \Delta_2$ for irrational $p \in (3/4, 1)$, and that $y_1^2 = 0 \in \mathbb{R}^2$.

It follows by definition of (AFP) at time $t = 2$ that $y_1^2 = y_1^2 = 0 \in \mathbb{R}^2$ and $y_2^2 = -A^\top x_1^1 = (-(2p-1), (2p-1))$, and that $x_1^2 = x_1^1 \in \Delta_2$ and $x_2^2 = (0, 1) \in \Delta_2$. Then, at $t = 3$, we further have $y_1^3 = y_1^2 + Ax_2^2 = (-1, 1)$ and $y_2^3 = y_2^2 - A^\top x_1^1 = 2 \cdot y_2^2$.

Then for $t \geq 3$, switching to the equivalent, lower-dimensional iterates $\{w^t\}$ and $\{z^t\}$, we can further directly compute (by definition of (AFP Dual)):

$$(t = 3) \quad \begin{cases} z^3 = (-1, -2(2p-1)) \in P_1 \\ \widetilde{z}^4 = (z_1^3, z_2^2) = (-1, -(2p-1)) \in P_1 \end{cases} \qquad w^3 = \mathrm{Q}(\widetilde{z}^4) = e_1$$

$$(t = 4) \quad \begin{cases} z^4 = z^3 + (-1, 1) = (-2, -4p+3) \in P_1 \\ \widetilde{z}^5 = (z_1^3, z_2^4) = (-1, -4p+3) \in P_1 \end{cases} \qquad w^4 = \mathrm{Q}(\widetilde{z}^5) = e_1$$

$$(t = 5) \quad \begin{cases} z^5 = z^4 + (-1, 1) = (-3, -4p+4) \in P_2 \\ \widetilde{z}^6 = (z_1^5, z_2^4) = (-3, -4p+3) \in P_1 \end{cases} \qquad w^5 = \mathrm{Q}(\widetilde{z}^6) = e_1$$

$$(t = 6) \quad \begin{cases} z^6 = z^5 + (-1, 1) = (-4, -4p+5) \in P_2 \\ \widetilde{z}^7 = (z_1^5, z_2^6) = (-3, -4p+5) \in P_2 \end{cases} \qquad w^6 = \mathrm{Q}(\widetilde{z}^7) = e_2.$$

Observe by Definition D.8 and the calculations above that Phase 1 begins at step $t_1 = 3$, and Phase 2 begins at phase $t_2 = 6$. Moreover, $\Delta\psi(z^5) = \Delta\psi(z^6) = 1$, meaning $\psi(z^{t_2}) - \psi(z^{t_1}) = 2 > 0$.

This strictly increasing energy growth between phases stems from the geometry of the predicted payoff vectors: in this instance, under Alternating Fictitious Play, when $z^t, z^{t-1} \in P_1$ and $z^t$ is near the boundary $P_{1 \sim 2}$, the predicted vector $\widetilde{z}^{t+1}$ always remains in $P_1$ and fails to "predict" the next region $P_2$. This results in strictly increasing energy growth when $z^{t+1} \in P_2$. This positive energy growth behavior near the boundary regions is the key difference between Alternating and Optimistic Fictitious Play (c.f., the invariants and energy growth claims of Lemma 4.10).

**Cycling invariants.** By continuing to compute the dual iterates $\{z^t\}$, we arrive at the following invariants, which establish a certain cycling behavior through the regions of $\widehat{\mathcal{P}}$. Specifically, it follows inductively that $z^{t-1}$ and $z^t$ must fall under one of the following cases (which subsequently determines $\widetilde{z}^{t+1}$, $z^{t+1}$, and the energy growth $\Delta\psi(z^t)$):

- **Case 1:** $z^{t-1}, z^t \in P_i$, and $z^t - z^{t-1} = S_i$.
  Then $\widetilde{z}^{t+1} \in P_i$, and either $z^{t+1} \in P_{i+1}$ with $\Delta\psi(z^t) = 1$, or $z^t \in P_i$ with $\Delta\psi(z^t) = 0$.

- **Case 2:** $z^{t-1} \in P_i$, $z^t \in P_{i+1}$, and $z^t = z^{t-1} + S_i$.
  Then $z^{t+1} \in P_{i+1}$, and either $\widetilde{z}^{t+1} \in P_i$ and $\Delta\psi(z^t) = 1$, or $\widetilde{z}^{t+1} \in P_{i+1}$ and $\Delta\psi(z^t) = 0$.

- **Case 3:** $z^{t-1} \in P_i$, $z^t \in P_{i \sim (i+1)}$ and $z^t = z^{t-1} + S_i$.
  If $\widetilde{z}^{t+1} \in P_i$, then $z^{t+1} \in P_{i+1}$ and $\Delta\psi(z^t) = 1$. If $\widetilde{z}^{t+1} \in P_{i \sim (i+1)}$, then also $z^{t+1} \in P_{i+1}$, with $\Delta\psi(z^t) \in \{0, 1\}$ depending on the tiebreaking of Q.

- **Case 4:** $z^{t-1} \in P_{i \sim (i+1)}$ and $z^t \in P_{i+1}$. If $\widetilde{z}^{t+1} \in P_{i \sim (i+1)}$, then $z^{t+1} \in P_{i+1}$, and $\Delta\psi(z^t) \in \{0, 1\}$ depending on the tiebreaking of Q. If $\widetilde{z}^{t+1} \in P_{i+1}$, then $z^{t+1} \in P_{i+1}$ and $\Delta\psi(z^t) = 0$.

Note that the cases above account for (a) the variability of $\widetilde{z}^{t+1}$ depending on the parity of $t$, and (b) any variability in $z^{t+1}$ depending on the tiebreaking decision encoded in Q. In summary, we deduce from the four cases above the following consequences:

1. Between phases, energy strictly increases in at most 2 iterations. By definition of the energy function $\psi$ under Matching Pennies, each one-step increase has magnitude 1, and thus $\psi(z^{t_k}) - \psi(z^{t_{k-1}}) \leq 2$, which proves claim (i) of the proposition.

2. Again using the definition of $\psi$ under Matching Pennies, we have $\psi(z^t) = \|z^t\|_1$. The cases above then imply that each $\tau_k = \|z^t\|_1 + c_k$ (for some aboslute constant $c_k$), and it follows that $\tau_k = \Theta(\psi(z^t))$, which proves claim (ii) of the proposition.

3. Finally, using the definition of $S$ under Matching Pennies, along with the fact that initially $z_1^2 = 0$, it holds that each $z_1^t$ is integral. Thus between regions $P_2$ and $P_3$, and between $P_4$ and $P_1$, one dual iterate will always lie on the boundary $P_{2\sim 3}$ or $P_{3\sim 4}$, respectively. In these cases, depending on the tiebreaking rule of Q, the change in energy may be zero when crossing between regions of $\mathcal{P}$. On the other hand, due to the initialization $x_1^1 = (p, 1-p)$ for irrational $p \in (3/4, 1)$, it follows for $t \geq 2$ that all $z_2^t$ are irrational. Thus no tiebreaking occurs when the dual iterates transition between regions $P_1$ and $P_2$ and between $P_3$ and $P_4$. Thus under transitions between these phases (which by symmetry amount for at least $\Omega(K/2)$ total phases), we have by the cases above that energy is strictly increasing by at least 1. This proves claim (iii) of the proposition. $\qquad\square$

# E   Additional Experimental Results

In this section, we provide more details on the experimental evaluations from Figure 1 and Section 5, and we also present additional experimental results. The goal of these experiments is to give further empirical evidence that the constant regret guarantee of Theorem 3.5 for two-strategy games also holds in higher dimensions.

## E.1   Details on Experimental Setup

First, we note that all code used to run experiments can be found in the supplementary material. In this paper, all experiments were run locally on a single personal computer.

**Families of payoff matrices.**   Aside from the (Matching Pennies) game, our experimental evaluations of Fictitious Play variants are performed on three high-dimensional families of payoff matrices:

- **Identity matrices:** Here, the payoff matrix is the $n \times n$ identity matrix $I_n$ (i.e., the diagonal matrix with diagonal entries all 1). Recall that for standard FP, Abernethy et al. (2021a) established an $O(\sqrt{T})$ regret bound using fixed lexicographical tiebreaking.

- **Generalized Rock-Paper-Scissors (RPS) matrices:** Here, the payoff matrix is the $n \times n$ generalization of the classic three-strategy Rock-Paper-Scissors game. Specifically, $A$ is the matrix with entries $A_{i,j}$ given by

$$A_{i,j} := \begin{cases} -1 & \text{if } j = i + 1 \pmod{n} \\ 1 & \text{if } j = i - 1 \pmod{n} \qquad \text{for all } i, j \in [n] \ . \\ 0 & \text{otherwise} \end{cases} \tag{40}$$

  For standard FP, Lazarsfeld et al. (2025) established an $O(\sqrt{T})$ regret bound for all such RPS matrices (using any tiebreaking rule), including when $A$ is scaled by a constant, and when the non-zero entries have non-uniform weights.

- **Random [0,1] matrices:** We also consider $n \times n$ payoff matrices with uniformly random entries in $[0, 1]$. For these payoff matrices, there are no existing $O(\sqrt{T})$ regret bounds for standard FP.

**Tiebreaking rules.**   To evaluate the robustness of regret guarantees to the tiebreaking method, we run the FP variants using both (a) fixed *lexicographical tiebreaking* (e.g., as in Abernethy et al. (2021a)) and (b) uniformly *random tiebreaking* (e.g., over the entries of the argmax set).

**Random initializations.**   To evaluate the robustness of regret guarantees to the players' initial strategies, we evaluated the Fictitious Play variants over multiple random initializations of $x_1^0, x_2^0 \in \Delta_n$ (for the Alternating FP initialization from Figure 1, note that the stated initialization is for

$x_1^1 \in \Delta_n$, as in the notation of Definition D.1). To generate a random initialization $x \in \Delta_n$, we sample $v \in [0,1]^n$ with independent, uniformly random entries, and normalize $x := v/\|v\|_1$.

## E.2 Empirical Regret Comparisons of Fictitious Play and Optimistic Fictitious Play

**Regret comparisons under randomized tiebreaking.** In Table 2 of Section 5, we presented regret comparisons of Optimistic FP and standard FP on the three families of payoff matrices described above in Section E.1, using *fixed lexicographical tiebreaking*. In Table 3, we show the results of an identical experimental setup, now using randomized tiebreaking. As in Table 2, the entries of Table 3 report average empirical regrets (and standard deviations) over 100 random initializations, where for each initialization, each algorithm was run for $T = 10000$ iterations.

| *Dimension*: | 15×15 | | 25×25 | | 50×50 | |
|:---:|:---:|:---:|:---:|:---:|:---:|:---:|
| Payoff Matrix | **FP** | **OFP** | **FP** | **OFP** | **FP** | **OFP** |
| **Identity** | $154.4 \pm 4.2$ | $8.4 \pm 1.7$ | $162.3 \pm 3.4$ | $12.9 \pm 1.6$ | $166.9 \pm 2.2$ | $25.0 \pm 2.3$ |
| **RPS** | $235.2 \pm 6.6$ | $2.8 \pm 0.5$ | $241.5 \pm 6.1$ | $3.2 \pm 0.9$ | $242.9 \pm 5.6$ | $2.6 \pm 0.8$ |
| **Random [0, 1]** | $93.4 \pm 5.0$ | $2.7 \pm 0.6$ | $137.1 \pm 6.1$ | $7.0 \pm 1.1$ | $176.2 \pm 6.3$ | $12.2 \pm 1.4$ |

Table 3: Empirical regret of FP and OFP using randomized tiebreaking. Each entry reports an average and standard deviation (over 100 random initializations) of total regret after $T = 10000$ steps.

As in Table 2, the results of Table 3 similarly show that Optimistic FP empirically obtains bounded regret compared to the roughly $O(\sqrt{T})$ regret of standard FP for each payoff matrix and dimension.

**Additional plots from fixed initializations.** To further compare the empirical regrets of standard FP and Optimistic FP, we present plots of the two algorithms run from fixed initializations, similar to Figure 1 from Section 1 (which also included a comparison with AFP). In each plot, we consider the three families of identity, RPS, and random matrices described earlier in Section E.1. Note in particular that for the RPS game (including in Figure 1 of Section 1), for better visual comparison with the other games, we use the payoff matrix specified in (40), but scaled by the constant $2/3$.

Figures 4, 5, and 6 show these comparisons for $15 \times 15$ and $25 \times 25$ matrices, using both randomized and lexicographical tiebreaking. In each instance, we again observe that Optimistic FP has bounded empirical regret compared to the $\sqrt{T}$ regret of standard FP.

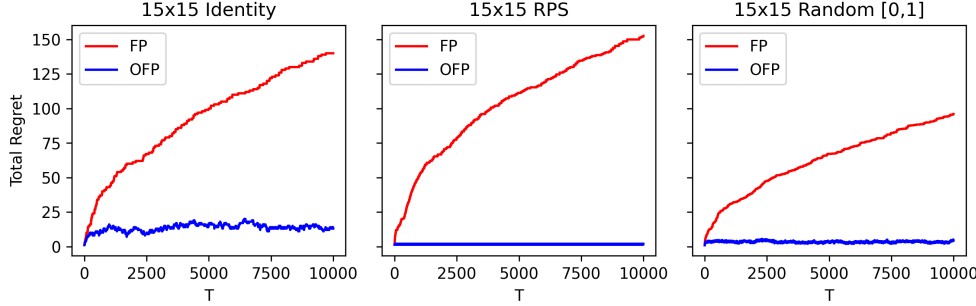

Figure 4: Empirical regret of standard FP and Optimistic FP (OFP) using **randomized tiebreaking** on three $15 \times 15$ payoff matrices. For each payoff matrix, each algorithm was initialized from $x_1^0 = e_1, x_2^0 = e_n$ and run for $T = 10000$ iterations.

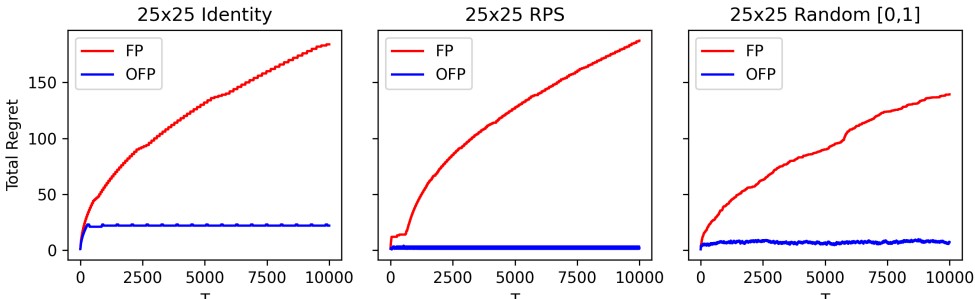

Figure 5: Empirical regret of standard FP and Optimistic FP (OFP) using **lexicographical tiebreaking** on three $25 \times 25$ payoff matrices. For each payoff matrix, each algorithm was initialized from $x_1^0 = e_1, x_2^0 = e_n$ and run for $T = 10000$ iterations.

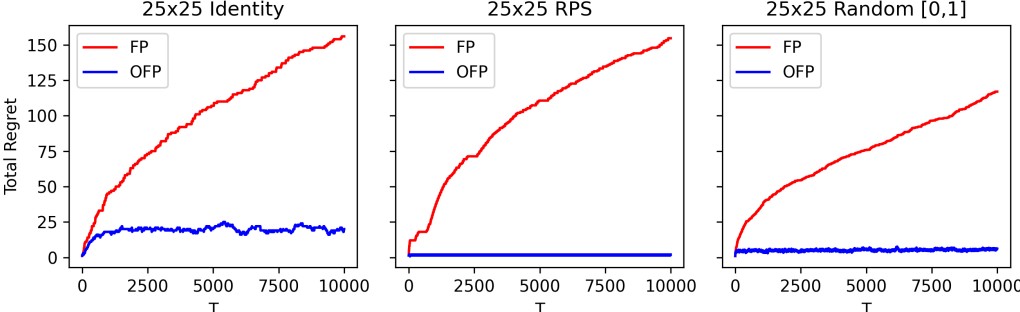

Figure 6: Empirical regret of standard FP and Optimistic FP (OFP) using **randomized tiebreaking** on three $25 \times 25$ payoff matrices. For each payoff matrix, each algorithm was initialized from $x_1^0 = e_1, x_2^0 = e_n$ and run for $T = 10000$ iterations.

