# OpenReview forum: "Optimism Without Regularization: Constant Regret in Zero-Sum Games"
_NeurIPS.cc/2025/Conference — NeurIPS 2025 poster_

### Official Review · Reviewer_kPJF · 2025-06-30

**Clarity:** 3
**Significance:** 4
**Originality:** 3
**Rating:** 5
**Confidence:** 4

**Summary:**

The paper demonstrates that constant regret can be achieved with Optimistic Fictitious Play (OFP) without regularisation in two-player zero-sum games, and the key contribution is identifying that constant regret is possible by associating primal variables with enumerated game states, represented by vertices of the joint simplex. The authors leverage geometric insights and a Lyapunov-esque energy-function framework to establish a constant (joint) regret bound. Experimental evidence is provided suggesting that OFP enjoys the same constant-regret properties in higher-dimensional games.

**Questions:**

* Upon reviewing the proof, I don't see any obvious obstacles precluding direct generalization to higher-dimensional games. Is the primary bottleneck in extending results from 2x2 to higher-dimensional settings solely related to a generalization of Assumption 1 and Propositions 4.1 and 4.2? Would addressing these issues require only a more careful analysis of the relevant subspace dimensions, or might it necessitate different mathematical tools?
* Do you have theoretical insights or intuitive explanations for why constant regret empirically holds in larger-dimensional games despite the theoretical analysis currently covering only 2×2 settings?
* In some practical cases, the ratio $a_{\max}/a_{\min}$ might significantly exceed the actual realized losses. Could a tighter regret bound potentially be derived, possibly depending on the trajectories observed or norms of the subset of the payoff matrix A realized during play? Might gradient-adaptive methods provide tighter bounds here, even though they typically incorporate regularisation?
* Regularised methods often exhibit greater robustness against perturbations in the payoff function. Given this, could the authors comment on how robust OFP is to noise or inaccuracies in the feedback or computation of the payoff matrix A?

**Ethical Concerns:**

["NO or VERY MINOR ethics concerns only"]

**Final Justification:**

The authors' have clearly answered my questions, and nothing significant is unresolved. I attribute a high weight to technical novelty, which this paper exhibits in the geometric insights developed throughout, the possibility of extension, interest of results to a broad audience, overall significance, and a clear delineation of open problems. This paper is very comfortably aligned with all these desiderata, and I am therefore happy to recommend its acceptance.

**Limitations:**

Yes.

**Paper Formatting Concerns:**

No concerns.

**Quality:**

4

**Strengths And Weaknesses:**

Strengths:
* The presented results constitute the first O(1) regret bound for fictitious play without regularisation. This is already a result of interest.
* The analysis utilizing energy functions is elegant, insightful, and affords some interesting tools to build upon, and potentially re-apply to the study of other approaches to the problem. In particular, I would foresee the energy function argument being applied to other related algorithmic approaches to two-player zero-sum games.
* While analyzing this simple algorithm, the authors provide a series of intriguing observations that elegantly distinguish optimistic fictitious play from standard fictitious play. The observation of the existence of the invariant and the analysis stemming from the game-state representation are particularly interesting.
* Experiments in higher-dimensional settings are compelling and strongly support the hypothesis that findings from the 2x2 setting extend to higher dimensions.
* The story of the paper is extremely clear, and the writing and communication of key ideas is concise, intuitive and easy to follow.

Weaknesses:
* Despite strong evidence that higher-dimensional generalizations are plausible, the paper’s rigorous theoretical results remain limited explicitly to 2x2 settings.
* The notation occasionally becomes cumbersome, making the proofs somewhat difficult to parse at certain points. However, given that the proof encompasses numerous intricate observations and ideas (and since it’s unclear how notation could be significantly simplified), this weakness is relatively minor.
* The simplicity of the presented method raises questions about its broader applicability in more generic or practical scenarios (see final question).

---

> ### Author Rebuttal · Authors · 2025-07-30
>
> Thank you for reviewing our work. We are glad you found the result interesting and clearly presented! Below we reply to some of the questions asked in the review:
>
> **On extending the analysis beyond 2x2 zero-sum games**
>
> We believe the core ideas and tools used to prove our main theorem will be useful for proving a more general result beyond 2x2 two-player zero-sum games. In short, we believe the key remaining challenge for many-action zero-sum games is to generalize the subspace dynamics machinery stemming from, e.g., Proposition 4.2 to the higher-dimensional regime, which will require additional insights.
>
> In more details: note that to prove the main theorem (Theorem 3.1), the analysis introduced in Section 4 is indeed somewhat specialized for the 2x2 case. However, this approach relies on the more general geometric intuition introduced in Section 3 (specifically, equation 5), which holds for all higher-dimensional two-player zero-sum games (see also the expanded derivation in Section A.3). Indeed, equation 5 implies that, for any general two-player zero-sum game, the one-step change in energy under OFP is always non-positive when the optimistic/predicted dual variable and the realized dual variable map to the same primal action under the update rule. The main technical lemma of Lemma 4.10 establishes sufficient invariants for when this alignment is true in the 2x2 case, and the proof leverages the subspace property (e.g., Proposition 4.2) for 2x2 zero-sum games with a unique, interior equilibrium.
>
> Based on our empirical results, we believe that similar invariants also hold for many-action two-player zero-sum games, and thus the high-level approach of our analysis (showing that the change in energy is always non-increasing if its magnitude ever exceeds some matrix-dependent constant) can be re-used. However, we believe that formally establishing such invariants in full generality will still require new insights, in particular due to the higher-dimensional nature of the dual iterates (e.g., the two-dimensional subspace property no longer holds). These new insights may also need to leverage structural properties of the underlying class of zero-sum game (for example, using properties of diagonal payoff matrices or skew-symmetric payoff matrices).
>
> We believe that developing these new insights will contribute more broadly to better understanding the performance of unregularized learning in games, an area which has had an exciting recent resurgence in the past year, and as mentioned by Reviewer qaZX, also has implications for practical equilibrium computation. To this end, we believe our result and proof technique for the 2x2 case serves as a useful starting point for the learning in games community to continue to make progress in this area.
>
> **Reply to amax/amin question**
>
> Yes, in general the constant amax/amin can be large, but this ratio essentially encodes the minimum-support mass in the Nash distribution (and note that constant regret and O(1/T) time-averaged convergence to Nash equilibrium essentially implies that at least a number of iterations inversely proportional to this magnitude must have occurred). It is possible that an even sharper (constant) bound can be derived in some instances, but in this work we did not attempt to optimize such constant dependencies.
>
> **Reply to noisy feedback question**
>
> While we did not investigate the robustness of Optimistic FP to perturbations in the payoffs or feedback vectors, we believe for (stochastically) unbiased noisy feedback that OFP may still have bounded energy / constant regret. Studying this formally would be an interesting direction for future work.

---

> ### Comment · Reviewer_kPJF · 2025-08-05
>
> Many thanks for the detailed response to my questions. Indeed, it has helped me appreciate the broader interest and significance of the results. I maintain my recommendation for its acceptance, and will increase my significance score.

---

### Official Review · Reviewer_qaZX · 2025-07-02

**Clarity:** 3
**Significance:** 3
**Originality:** 3
**Rating:** 5
**Confidence:** 4

**Summary:**

This work studies the performance of unregularized online learning (the fictitious play algorithm aka FTL) in two-player zero sum games.  More specifically, they analyze FTL with two classic modifications (optimism and alternation) known to improve regret performance of regularized learning algorithms in games, and establish a positive result for the former and a negative result for the latter when regularization is removed, specifically in two-action games such as matching pennies.

**Questions:**

Could the authors clarify where the potential approach fails for many-action games?

**Ethical Concerns:**

["NO or VERY MINOR ethics concerns only"]

**Final Justification:**

I stand by my positive assessment.

**Limitations:**

Yes

**Quality:**

4

**Strengths And Weaknesses:**

The positive result for optimistic FTL, though constrained to two-action games, I believe to be meaningful progress in the study of unregularized learning in games, which still has major open problems such as the weak fictitious play conjecture.  It is a genre of algorithms that is worth understanding due to their per-iterate time efficiency in solving large two-player zero-sum games in practice where the cost of computing a regularized best response is potentially high.  Also, the separation demonstrated from the performance of Alternating FTL gives grounds to the fact that optimism is perhaps the more natural modification to vanilla online learning in games.

This result is also interesting as it is somewhat unexpected to me.  The O(1) regret in two-player zero-sum games due to Syrgkanis et. al. comes from the RVU regret bound.  In the worst case, this regret bound makes no improvement over the adversarial $O(\sqrt{T})$ rate, as to be expected due to the existence of simple Bernoulli adversaries forcing $\Omega(\sqrt{T})$.  However, with the added structure that the loss sequence experienced by the learner is coming from the learning of opponents in a game, we have that this loss sequence is *smooth*.  This smoothness, when plugged into the RVU bound, gives the improved $o(\sqrt{T})$, even $O(1)$ sum of regret bound, which is sufficient for convergence to equilibrium in zero-sum games, unlike general sum games.  Thus, what astonishes me is that you can also get the matching $O(1)$ regret bound in this setting where non-regularized algorithms are inherently not smooth, having effectively infinite step size and always selecting actions at the boundary.  Accordingly, the proof technique of this result must be built in a fundamentally different way than RVU.  I find the potential argument the authors utilize instead to be quite satisfying.

The main drawback of this work is in its limitation to two-player zero-sum games, and more so *two-action* two-player zero-sum games.  As with my earlier skepticism, I find it unlikely that you will have a positive ($O(1)$ regret) result for unregularized learning in multi-player general-sum games as these learners lack smoothness.  Also, the potential argument as presented here does not seem to have hope to extend to many action two-player zero-sum games.  That said, the class of two-action two-player zero-sum games is sufficiently rich to establish learning-in-games lower bounds for even regularized algorithms without optimism (i.e. MWU having $\Omega(\sqrt{T})$ regret in matching pennies a la Chen and Peng).  So, the fact that you get a positive result for even matching pennies, as long as you are using optimism, even without regularization, I find to be a meaningful result standing on its own when juxtaposed to the Chen Peng result.  Thus, I recommend the acceptance of this result despite the aforementioned limitations, and perhaps the publicization of this result will encourage others to attempt to tackle the unregularized, optimistic, many action, two player zero sum setting.  A positive result there could be a potential boon for practical equilibrium computation.

---

> ### Author Rebuttal · Authors · 2025-07-30
>
> Thank you for reviewing our work. We are glad you found the main result meaningful and surprising, and we agree that our results can help set the stage for interesting followups and more progress in the area! Below we reply to the question asked in the review:
>
> **On extending the analysis beyond 2x2 zero-sum games**
>
> We believe the core ideas and tools used to prove our main theorem will be useful for proving a more general result beyond 2x2 two-player zero-sum games. In short, we believe the key remaining challenge for many-action zero-sum games is to generalize the subspace dynamics machinery stemming from, e.g., Proposition 4.2 to the higher-dimensional regime, which will require additional insights.
>
> In more detail: note that to prove the main theorem (Theorem 3.1), the analysis introduced in Section 4 is indeed somewhat specialized for the 2x2 case. However, this approach relies on the more general geometric intuition introduced in Section 3 (specifically, equation 5), which holds for all higher-dimensional two-player zero-sum games (see also the expanded derivation in Section A.3). Indeed, equation 5 implies that, for any general two-player zero-sum game, the one-step change in energy under OFP is always non-positive when the optimistic/predicted dual variable and the realized dual variable map to the same primal action under the update rule. The main technical lemma of Lemma 4.10 establishes sufficient invariants for when this alignment is true in the 2x2 case, and the proof leverages the subspace property (e.g., Proposition 4.2) for 2x2 zero-sum games with a unique, interior equilibrium.
>
> Based on our empirical results, we believe that similar invariants also hold for many-action two-player zero-sum games, and thus the high-level approach of our analysis (showing that the change in energy is always non-increasing if its magnitude ever exceeds some matrix-dependent constant) can be re-used. However, we believe that formally establishing such invariants in full generality will still require new insights, in particular due to the higher-dimensional nature of the dual iterates (e.g., the two-dimensional subspace property no longer holds). These new insights may also need to leverage structural properties of the underlying class of zero-sum game (for example, using properties of diagonal payoff matrices or skew-symmetric payoff matrices).
>
> To reiterate, we believe that developing these new insights will contribute more broadly to better understanding the performance of unregularized learning in games, an area which has had an exciting recent resurgence in the past year, and as you mention, also has implications for practical equilibrium computation. To this end, we believe our result and proof technique for the 2x2 case serves as a useful starting point for the learning in games community to continue to make progress in this area.

---

> > ### Comment · Reviewer_qaZX · 2025-08-07
> >
> > Thanks for your reply.

---

### Official Review · Reviewer_ySgf · 2025-07-02

**Clarity:** 3
**Significance:** 3
**Originality:** 2
**Rating:** 4
**Confidence:** 3

**Summary:**

This paper studies the regret of optimistic fictitious play in $2 \times 2$ zero-sum games. The authors show that the algorithm achieves constant regret without requiring any regularization. The main technical contribution lies in proving that an energy function (or potential function) remains bounded by an absolute constant. Numerical simulations are provided to support the theoretical findings.

**Questions:**

Can the authors more clearly state the fundamental challenges in extending their result beyond the $2 \times 2$ setting, and discuss any potential directions or ideas for addressing these challenges?

**Ethical Concerns:**

["NO or VERY MINOR ethics concerns only"]

**Final Justification:**

I am maintaining my score and my support for the acceptance of this work.

**Limitations:**

The main limitation is that the results work only for $2 \times 2$ games.

**Quality:**

3

**Strengths And Weaknesses:**

Strengths:

- The paper is very well-written. The contributions are clearly stated, the results and simulations are nicely presented, and the technical approach is easy to follow.

- Zero-sum games are a benchmark setting in game theory, and fictitious play is one of the most classical algorithms. However, understanding the convergence rate of the average iterate (or regret) has remained a long-standing open problem (e.g., Karlin’s weak conjecture). Showing constant regret—which implies a $\mathcal{O}(1/T)$ convergence rate for the average iterate—using a variant of fictitious play is a significant and meaningful contribution.

Weaknesses:

(1) The main result is limited to $2 \times 2$ games, as acknowledged by the authors. While this is a natural first step, it limits the generality of the findings.

(2) Since similar energy functions have already been constructed in prior work, the technical novelties of this paper are somewhat unclear.

---

> ### Author Rebuttal · Authors · 2025-07-30
>
> Thank you for reviewing our work. We are happy you found our contribution significant and our technical approach easy to follow! Below are replies to two points made in the review:
>
> **On the technical novelty of the paper.**
>
> Note that the energy function used in the analysis is equivalent to the scaled duality gap, which is a fundamental quantity of study for zero-sum games (see, e.g., Proposition 2.1 and Proposition 3.4). The technical novelty of the paper is in proving for Optimistic FP that the energy is bounded by a constant (leading to constant regret, and O(1/T) time-average convergence to Nash). This proof of bounded energy for Optimistic FP relies on new geometric insights that are different from the techniques of prior works that show \sqrt{T} regret bounds for standard Fictitious Play (e.g., in [Abernethy et al., 2021a] and [Lazarsfeld et al., 2025]) where the energy is not bounded but in fact growing).
>
> **On extending the analysis beyond 2x2 zero-sum games**
>
> We believe the core ideas and tools used to prove our main theorem will be useful for proving a more general result beyond 2x2 two-player zero-sum games. In short, we believe the key remaining challenge for many-action zero-sum games is to generalize the subspace dynamics machinery stemming from, e.g., Proposition 4.2 to the higher-dimensional regime, which will require additional insights.
>
> In more detail: note that to prove the main theorem (Theorem 3.1), the analysis introduced in Section 4 is indeed somewhat specialized for the 2x2 case. However, this approach relies on the more general geometric intuition introduced in Section 3 (specifically, equation 5), which holds for all higher-dimensional two-player zero-sum games (see also the expanded derivation in Section A.3). Indeed, equation 5 implies that, for any general two-player zero-sum game, the one-step change in energy under OFP is always non-positive when the optimistic/predicted dual variable and the realized dual variable map to the same primal action under the update rule. The main technical lemma of Lemma 4.10 establishes sufficient invariants for when this alignment is true in the 2x2 case, and the proof leverages the subspace property (e.g., Proposition 4.2) for 2x2 zero-sum games with a unique, interior equilibrium.
>
> Based on our empirical results, we believe that similar invariants also hold for many-action two-player zero-sum games, and thus the high-level approach of our analysis (showing that the change in energy is always non-increasing if its magnitude ever exceeds some matrix-dependent constant) can be re-used. However, we believe that formally establishing such invariants in full generality will still require new insights, in particular due to the higher-dimensional nature of the dual iterates (e.g., the two-dimensional subspace property no longer holds). These new insights may also need to leverage structural properties of the underlying class of zero-sum game (for example, using properties of diagonal payoff matrices or skew-symmetric payoff matrices).
>
> We believe that developing these new insights will contribute more broadly to better understanding the performance of unregularized learning in games, an area which has had an exciting recent resurgence in the past year, and as mentioned by Reviewer qaZX, also has implications for practical equilibrium computation. To this end, we believe our result and proof technique for the 2x2 case serves as a useful starting point for the learning in games community to continue to make progress in this area.

---

> > ### Comment · Reviewer_ySgf · 2025-08-05
> > **Official Comment by Reviewer ySgf**
> >
> > Thank the authors for the response. I remain positive about the submission and will maintain my score.

---

### Decision · Program_Chairs · 2025-09-17

**Decision:**

Accept (poster)

**Comment:**

This paper shows that in 2x2 zero-sum games optimistic fictitious play constant regret without any regularization, While this result is narrow, it represents a nice and perhaps surprising contribution to our theoretical understanding of learning in this space.  Based on the discussions with the authors, some of the reviewers feel (and I agree) that the paper stands a good chance of providing inspiration toward further development in this space and that the novel techniques it develops may be useful in that.